# Perceptual coupling and decoupling of the default mode network during mind-wandering and reading

**Meichao Zhang[1]\*, Boris C Bernhardt[2], Xiuyi Wang[1], Dominika Varga[1], Katya Krieger-Redwood[1], Jessica Royer[2], Raúl Rodríguez-Cruces[2], Reinder Vos de Wael[2], Daniel S Margulies[3], Jonathan Smallwood[4], Elizabeth Jefferies[1]\***

[1]Department of Psychology, University of York, York, United Kingdom; [2]McConnell Brain Imaging Centre, Montreal Neurological Institute and Hospital, McGill University, Montreal, Canada; [3]Integrative Neuroscience and Cognition Centre (UMR 8002), Centre National de la Recherche Scientifique (CNRS) and Université de Paris, Paris, France; [4]Department of Psychology, Queen's University, Ontario, Canada

**Abstract** While reading, our mind can wander to unrelated autobiographical information, creating a perceptually decoupled state detrimental to narrative comprehension. To understand how this mind-wandering state emerges, we asked whether retrieving autobiographical content necessitates functional disengagement from visual input. In Experiment 1, brain activity was recorded using functional magnetic resonance imaging (fMRI) in an experimental situation mimicking naturally occurring mind-wandering, allowing us to precisely delineate neural regions involved in memory and reading. Individuals read expository texts and ignored personally relevant autobiographical memories, as well as the opposite situation. Medial regions of the default mode network (DMN) were recruited during memory retrieval. In contrast, left temporal and lateral prefrontal regions of the DMN, as well as ventral visual cortex, were recruited when reading for comprehension. Experiment two used functional connectivity both at rest and during tasks to establish that (i) DMN regions linked to memory are more functionally decoupled from regions of ventral visual cortex than regions in the same network engaged when reading; and (ii) individuals with more self-generated mental contents and poorer comprehension, while reading in the lab, showed more decoupling between visually connected DMN sites important for reading and primary visual cortex. A similar pattern of connectivity was found in Experiment 1, with greater coupling between this DMN site and visual cortex when participants reported greater focus on reading in the face of conflict from autobiographical memory cues; moreover, the retrieval of personally relevant memories increased the decoupling of these sites. These converging data suggest we lose track of the narrative when our minds wander because generating autobiographical mental content relies on cortical regions within the DMN which are functionally decoupled from ventral visual regions engaged during reading.

## Editor's evaluation

This manuscript is of broad interest to those interested in the relationship between mind-wandering and reading, at the behavioural and neural levels, including when both processes occur at the same time. As such, this manuscript has important implications for clarifying how the experience of mind-wandering while reading may occur.

**\*For correspondence:**
meichao.zhang@york.ac.uk (MZ);
beth.jefferies@york.ac.uk (EJ)

**Competing interest:** The authors declare that no competing interests exist.

**eLife digest** As your eyes scan these words, you may be thinking about what to make for dinner, how to address an unexpected hurdle at work, or how many emails are sitting, unread, in your inbox. This type of mind-wandering disrupts our focus and limits how much information we comprehend, whilst also being conducive to creative thinking and problem-solving.

Despite being an everyday occurrence, exactly how our mind wanders remains elusive. One possible explanation is that the brain disengages from visual information from the external world and turns its attention inwards. A greater understanding of which neural circuits are involved in this process could reveal insights about focus, attention, and reading comprehension.

Here, Zhang et al. investigated whether the brain becomes disengaged from visual input when our mind wanders while reading. Recalling personal events was used as a proxy for mind-wandering. Brain activity was recorded as participants were shown written statements; sometimes these were preceded by cues to personal memories. People were asked to focus on reading the statements or they were instructed to concentrate on their memories while ignoring the text. The analyses showed that recalling memories and reading stimulated distinct parts of the brain, which were in direct competition during mind-wandering.

Further work examined how these regions were functionally connected. In individuals who remained focused on reading despite memory cues, the areas activated by reading showed strong links to the visual cortex. Conversely, these reading-related areas became 'decoupled' from visual processing centres in people who were focusing more on their internal thoughts.

These results shed light on why we lose track of what we are reading when our mind wanders: recalling personal memories activates certain brain areas which are functionally decoupled from the regions involved in processing external information – such as the words on a page.

In summary, the work by Zhang et al. builds a mechanistic understanding of mind-wandering, a natural feature of our daily brain activity. These insights may help to inform future interventions in education to improve reading, comprehension and focus.

## Introduction

The human mind is remarkably flexible, capable of shifting focus from information in the external environment to perceptually decoupled states that are generated from information in memory (*Baird et al., 2014*; *Smallwood, 2013*; *Smallwood et al., 2021b*). This capacity for self-generating mental content is ubiquitous across cultures and has links to both beneficial and detrimental features of cognition (*Mooneyham and Schooler, 2013*). Although self-generated states are common in daily life (*Kane et al., 2007*; *Poerio et al., 2013*), they can be problematic if they occur during reading (*McVay and Kane, 2012*; *Smallwood et al., 2008*). The detrimental effects of mind-wandering on reading are believed to occur because this state elicits perceptual decoupling that disrupts narrative comprehension (*Smallwood, 2011*; *Smallwood, 2013*).

Contemporary work in cognitive neuroscience has shown that both reading for comprehension (*Andrews-Hanna et al., 2014b*; *Binder et al., 2009*; *Mar, 2011*; *Yang et al., 2019*), and off-task states (*Christoff et al., 2009*; *Konu et al., 2020*), are linked to activity within the default mode network (DMN). For example, the building blocks of reading for comprehension are conceptual representations supported by regions overlapping with DMN in the anterior, ventral and lateral temporal lobe (*Lambon Ralph et al., 2017*). In contrast, the content of self-generated thoughts often comes from autobiographical memory (*Smallwood et al., 2011*), linked to medial regions of the DMN including posterior cingulate, ventral prefrontal and inferior parietal cortex (*Ritchey and Cooper, 2020*). Furthermore, studies of individual differences highlight that both better reading comprehension, as well as greater tendency for off-task thought, are predictable based on neural patterns in the DMN, as well as in other cortical regions (*Smallwood et al., 2013*; *Zhang et al., 2019*). For example, individuals who are better at reading for comprehension show more functional integration between lateral and medial elements of the DMN, while individuals who tend to be more off-task show greater decoupling between the DMN and regions of visual cortex important for visual processing during reading (*Smallwood et al., 2013*; *Zhang et al., 2019*).

Converging empirical and theoretical evidence, therefore, suggests that both reading for comprehension, and off-task self-generated states, depend on regions within the broader DMN. Recent views suggest the DMN's role in cognition emerges from this system's topographic location, with core nodes located in regions that are distant in both functional and structural terms from unimodal cortex (*Margulies et al., 2016*). This topographical organisation has been argued to explain the role of the DMN in multiple cognitive states because it locates this network at the end of information processing streams that are necessary for relatively abstract tasks (like reading comprehension) but also explains why the same network can be involved in states that require disengagement from the here and now (such as mind-wandering; *Smallwood et al., 2021a*). This topographically informed view of the DMN provides a novel hypothesis for why mind-wandering creates a situation in which we lose track of the meaning of the words we are reading: the process of generating mental content using information from memory leads to a perceptually decoupled state associated with poor comprehension. When this occurs, there is a shift in the balance of neural activity within the DMN, away from DMN regions functionally coupled with ventral visual regions important for reading comprehension, to other regions within DMN that are more isolated from perceptual input.

To test this account of the consequences of mind-wandering while reading, we conducted two experiments using fMRI to measure brain activity. In Experiment 1, participants (N = 29) performed tasks that mimicked the experience of mind-wandering while reading. In one condition, participants were presented with information from an expository text on the screen and were asked to scan these words while instead retrieving a personally-relevant autobiographical memory. In a second condition, participants focused on a similar expository text, while refraining from attending to autobiographical information. Our experiment exploits the fact that self-generated states can be understood as the spontaneous engagement of processes that can also be recruited as part of a task (*Smallwood, 2013*). By exploiting this 'component process' view we could identify, with a high degree of precision, the neural regions involved in the two cognitive components of interest: autobiographical memory retrieval and reading for comprehension. Experiment 2 examined the functional architecture of regions involved in these two states, evaluating whether the DMN regions important for reading are more functionally connected to the ventral visual stream than the DMN regions involved in autobiographical memory (N = 243). Finally, we also sought to generalise these results by examining whether functional relationships between regions implicated in our experimental task, which was designed to mimic mind-wandering while reading, could predict individual differences in a previously published dataset that looked at naturally occurring mind-wandering during reading (N = 69; *Zhang et al., 2019*).

Foreshadowing our results, we found that (a) left lateral prefrontal and temporal regions of the DMN, within the dorsomedial subsystem (*Andrews-Hanna et al., 2010*; *Yeo et al., 2011*), are important for reading, while regions of the core DMN are engaged during autobiographical memory (medial prefrontal, posterior cingulate, and angular gyrus); (b) regions of the DMN linked to reading are more functionally connected to ventral visual regions than DMN regions implicated in autobiographical memory; (c) DMN regions linked to reading are decoupled from primary visual cortex for individuals who naturally generate more mind-wandering content during reading, generalising our experimental paradigm to an ecologically valid situation; and (d) this perceptual decoupling pattern linked to individual variation in mind-wandering during reading is also modulated by our task manipulation, with greater perceptual decoupling during the retrieval of personally relevant memories in the absence of sentences and less decoupling when participants remain focussed on reading despite distracting autobiographical memory cues. Together these data support the hypothesis that processes involved in the generation of mental content from memory depend on regions in the DMN that are functionally decoupled from ventral visual regions important for reading. Our analysis, therefore, suggests that the reason we lose track of the narrative when our mind wanders during reading is because the generation of autobiographical content relies on neural activity in core DMN regions and this encourages a state of perceptual decoupling that reduces comprehension of external input. This perceptually decoupled state is broadly comparable across a laboratory situation that mimics mind-wandering and an individual differences analysis of this feature of cognition in a more naturalistic setting.

## Results

### Experiment 1

#### Design

The experiment took place over 2 days. On Day 1, participants were asked to identify specific personal events linked to each autobiographical memory cue (words like PARTY). On Day 2, they recalled these memories when presented with the cue word in the scanner, and also completed the reading task, which involved reading factual sentences about similar concepts. For the task inside the scanner, we employed a within-subjects 2 × 2 design manipulating task (Autobiographical memory recall vs. Reading) and conflict (Conflict vs. No conflict). Participants were asked to either comprehend sentences presented word-by-word, or recall their personal memories, as a proxy for mind-wandering. To mimic the experience of mind-wandering while reading, in conflict trials, the two processes (Reading vs. Autobiographical memory recall) were pitted against one another – with participants required to

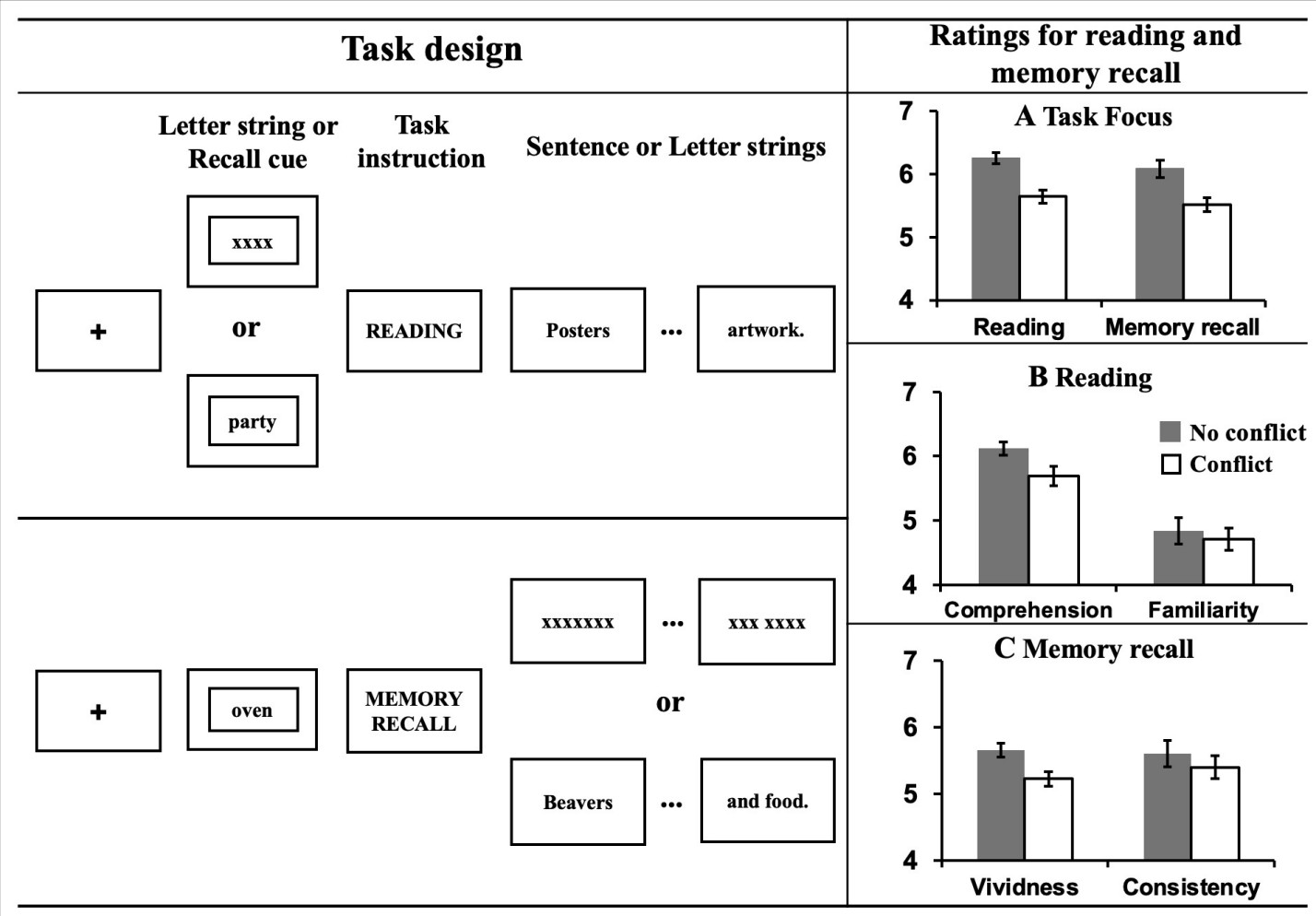

**Figure 1.** Task illustration and results of ratings. **Left hand panel: Task design of Experiment 1.** Using a counterbalanced design, participants either engaged in normal reading, or instead were focused on retrieval of personally-relevant information from memory. To mimic the mind-wandering while reading state, task conflict was created by presenting sentences during memory recall, or memory cues before the presentation of sentences. To understand the effect of meaningful input on memory retrieval, on some occasions 'X's were presented instead of text. To understand the effect of memory retrieval on reading, sometimes no memory was cued at the start of the reading trial. **Right hand panel: Evidence of mutual inhibition between reading and autobiographical memory retrieval.** (**A**) Participants rated their task focus as lower when reading while retrieving autobiographical memories (as well as vice versa). (**B**) Participants rated their comprehension of written material as lower when also retrieving autobiographical information. There was no effect on participants' ratings of their familiarity with the content of the sentences. (**C**) Participants rated their autobiographical memories as less vivid and less consistent with their previously generated memories when meaningful text was presented at the same time. Error bars show standard error of the mean (SEM).

either (1) recall autobiographical information, whilst irrelevant sentences were presented word-by-word on the screen or (2) read a sentence while trying to suppress a cued autobiographical memory. In this way, the conflict between these two opposing mental states was created by presenting sentences during memory recall, or memory cues before sentences, eliciting the kinds of cognitive states that occur when our minds wander during reading. To understand the conflict effect on each task, no conflict trials replaced sentences and autobiographical memory cue with letter strings (e.g. XXX); consequently, there was no conflicting information for participants to process (see the left hand of *Figure 1* for an illustration of the task). After each reading or memory recall trial, task focus ratings on a scale of 1 (i.e., not at all) to 7 (i.e., very much) were collected to index the extent to which participants were able to focus on the primary task.

## Behavioural results

Our first goal was to establish whether our experimental situation successfully mimicked features of mind-wandering while reading, namely (i) a focus on personally relevant information accompanied by (ii) a reduced focus on the text. A repeated-measures Analysis of Variance (ANOVA) was used to examine this question by comparing the effects of autobiographical memory retrieval on reading (and vice versa). Participants reported reduced task focus on the primary task for both reading and autobiographical memory retrieval when both tasks were presented at the same time (i.e., in Conflict conditions), $F(1,28) = 44.28$, $p < 0.001$, $\eta_p^2 = .61$ (see panel A in the right hand of *Figure 1*). There was no interaction between Task and Conflict, $F(1,28) = .12$, $p = 0.73$, $\eta_p^2 = .004$, indicating that the effect of conflict between tasks had an equivalent effect on mental focus in reading and autobiographical memory. For the reading trials, retrieval of autobiographical memories reduced rated comprehension, $F(1,28) = 10.40$, $p = 0.003$, $\eta_p^2 = .27$, but not participants' level of prior familiarity with the sentence material, $F(1,28) = 2.10$, $p = 0.16$, $\eta_p^2 = .07$ (see panel B in the right hand of *Figure 1*). For autobiographical memory, concurrent presentation of meaningful text reduced the vividness of the memories that were retrieved, $F(1,28) = 36.29$, $p < 0.001$, $\eta_p^2 = .56$, as well as the rated consistency between retrieval in the scanner and the memory described for the cue word outside the scanner, $F(1,28) = 8.19$, $p = 0.008$, $\eta_p^2 = .23$ (see panel C in the right hand of *Figure 1*). This pattern establishes that our paradigm successfully captures important features of the mind-wandering reading state, particularly the mutual inhibition between self-generated mental content and effective reading for comprehension.

## Neuroimaging results

Having established the expected pattern of competition between autobiographical memory retrieval and reading, we next considered the neural correlates that distinguish these states. To examine the key differences between reading and autobiographical memory in brain activity, we performed a univariate analysis using a general linear model (GLM) to identify (i) neural differences across these tasks (performing contrasts between the main effects of task condition) and (ii) parametric effects of task focus – that is regions in which brain activity correlated with rated focus on each trial. In this model, each of the four experimental conditions (i.e., *Reading with no conflict, Reading with conflict, Autobiographical memory retrieval with no conflict,* and *Autobiographical memory retrieval with conflict*) were included as Explanatory Variables (EV) of interest, along with the parametric effect of Task Focus for each condition. All the reading and memory recall trials were included in the analysis – including the rare catch trials with colour changes, since there were no behavioural differences in colour-change detection rates across these conditions (see *Appendix 1—figure 1*) and these catch trials were explicitly modelled in our analysis.

Contrasting reading with autobiographical memory retrieval highlighted a set of left lateralised regions within the temporal lobe and prefrontal cortex, including inferior frontal gyrus and superior and middle temporal gyri, which are recruited during reading. Activation in bilateral ventral visual cortex was also observed. In contrast, periods when autobiographical memory was the primary task were associated with greater neural activity in regions including medial and lateral prefrontal cortex, posterior cingulate cortex and angular gyrus. In *Figure 2A*, regions showing greater activity during reading are presented in warmer colours, and regions showing greater activity during autobiographical memory retrieval are presented in cooler colours. To confirm the most likely functional associations with these regions, we performed a meta-analysis using Neurosynth (see Materials and

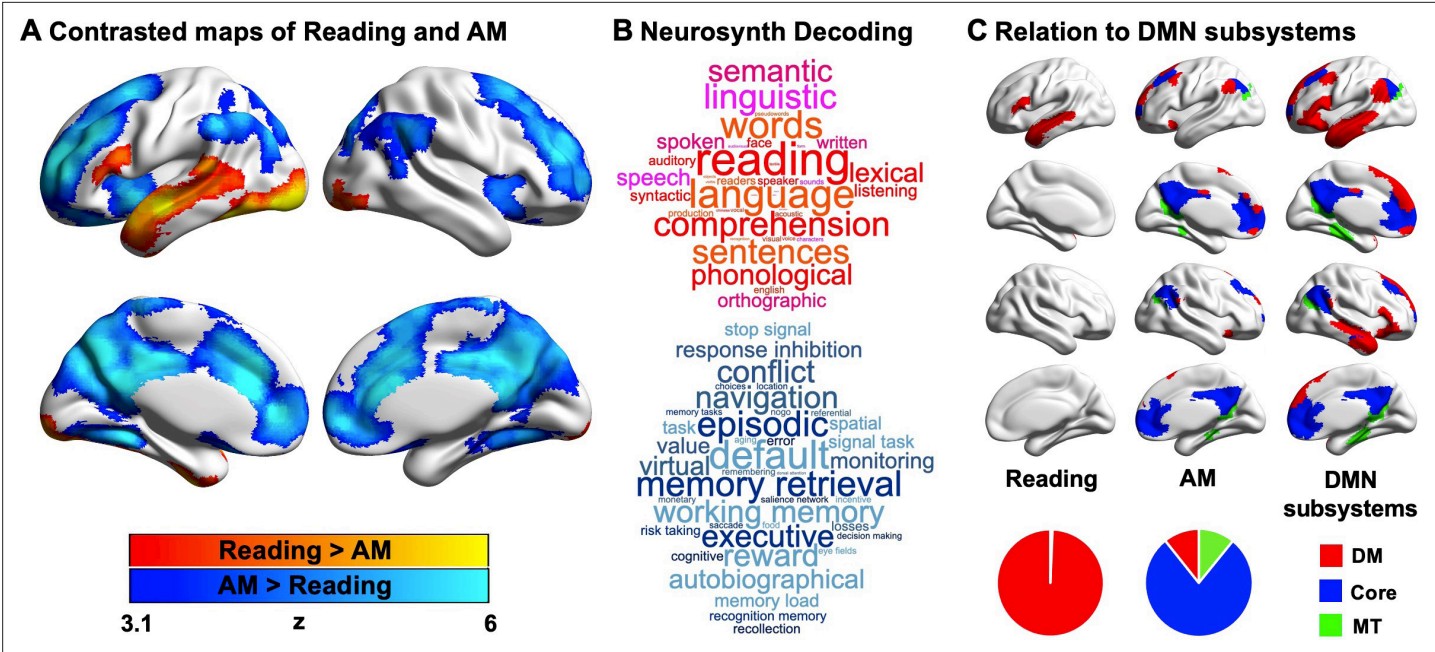

**Figure 2.** Neural activity associated with reading and autobiographical memory (AM) retrieval. (**A**) A comparison of regions showing significantly greater activity during reading (red) or autobiographical memory retrieval (blue). (**B**) A meta-analysis of the regions showing activity during reading and autobiographical memory retrieval using Neurosynth. In these word clouds, the font size of the item illustrates its importance and the colour indicates its association (red = reading, blue = autobiographical memory retrieval). (**C**) Relationship between the patterns of observed activity during reading and autobiographical memory retrieval and their relationship to the subsystems of the DMN as described by *Yeo et al., 2011*. In this panel, regions in red fall within the dorsomedial (DM) subsystem, regions in blue fall within the core subsystem and regions in green fall within the medial temporal (MT) subsystem. The pie charts show the proportion of significant voxels associated with each condition that fall within each subsystem. (Source data of unthresholded task contrasted maps, which are also used for Neurosynth decoding analysis in (**B**), could be found in Neurovault at https://neurovault.org/collections/9432/; *Figure 2—source data 1*).

The online version of this article includes the following source data for figure 2:

**Source data 1.** Task-activated voxels within each DMN subsystem.

methods). The results of this analysis are presented in *Figure 2B* in the form of word clouds where the font size describes the strength of the relationship and the colour describes the associated state (Red = reading, Blue = autobiographical memory retrieval). As expected, there was a correspondence between the psychological features of our conditions and the functional terms revealed by the meta-analysis, with regions linked to reading associated with terms such as 'reading' and 'language', while regions linked to autobiographical memory retrieval associated with terms like 'memory retrieval' and 'episodic'.

Prior studies have linked both semantic and autobiographical memory processes to the broader DMN (e.g. *Andrews-Hanna et al., 2014b*; *Lambon Ralph et al., 2017*; *Ritchey and Cooper, 2020*; *Sormaz et al., 2017*; *Spreng et al., 2009*; *Yang et al., 2019*), and we examined how the neural patterns associated with our states reflected the activation of different subsystems of the DMN (*Andrews-Hanna et al., 2010*; *Yeo et al., 2011*). For each of the significant DMN voxels associated with reading or autobiographical memory retrieval, we examined whether they fell within the dorsomedial, core or medial temporal DMN subsystems, as defined by *Yeo et al., 2011*. The results of this analysis are presented in *Figure 2C*, where the different columns show the different states (reading and autobiographical memory retrieval), and the different colours correspond to the DMN subsystems. The percentages of voxels falling within each subsystem are presented as pie charts at the foot of this panel. It can be seen that the DMN regions engaged during reading were entirely within the dorsomedial system (red; 100%), while the majority of the DMN voxels showing higher activity during autobiographical memory retrieval fell within the core subsystem (blue; 78%), with equal percentages in the dorsomedial (red; 11%) and medial temporal subsystems (green; 11%).

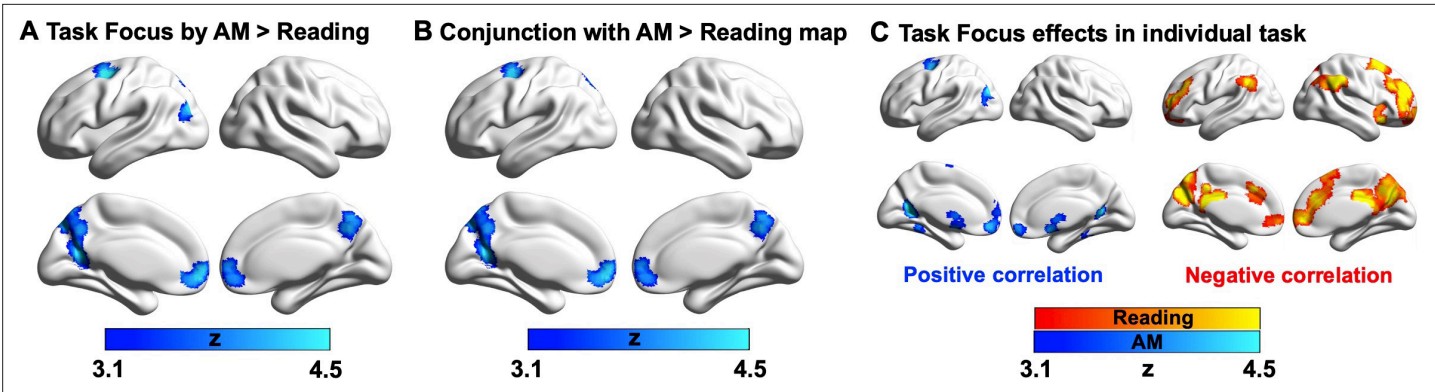

**Figure 3.** Parametric effects of task focus. (**A**) Regions showing a differential relationship with task focus across the two states (autobiographical memory (AM) retrieval and reading). These regions show greater activity when participants reported better focus on the task during autobiographical memory retrieval and poorer task focus during reading. (**B**) A formal conjunction between these regions in (**A**) showing more activation with greater task focus on memory retrieval and those showing greater activity during memory retrieval than reading (i.e., blue regions in *Figure 2A* showing main effect of task contrast). (**C**) Task focus effects in reading (red; negative correlation with task focus) and autobiographical memory recall task (blue; positive correlation with task focus), relative to the implicit baseline (i.e., the first fixation interval). (Source data for the unthresholded maps of task focus in **A-C** are provided in Neurovault at https://neurovault.org/collections/9432/).

Having established that autobiographical memory retrieval and reading activate distinct subsystems within the broader DMN (despite some overlap in brain activation elicited by these states relative to the letter string baseline, see *Appendix 1—figure 2*), we next explored the functional consequences of neural activity in these conditions. Our behavioural analysis demonstrated a pattern of mutual competition between the two conditions (*Figure 1*); we therefore examined the relationship between the observed pattern of neural activity in each condition and a persons' reported focus on the primary task. The results of this analysis are presented in *Figure 3A* where it can be seen that regions in medial prefrontal and parietal cortex, superior frontal gyrus and left lateral parietal cortex showed a stronger effect of task focus for autobiographical memory retrieval, relative to reading. We conducted a formal conjunction to identify how these regions mapped onto those showing differential activity for the two states. *Figure 3B* shows that clusters linked to better task focus for autobiographical memory retrieval also showed stronger activity during memory retrieval and lower activity during reading. Importantly, we confirmed that these regions showed significant associations with task focus separately in each task (see *Figure 3C*). Greater task focus during memory retrieval was associated with increased activation in medial prefrontal cortex, posterior parietal cortex (bordering dorsal lateral occipital cortex), superior frontal gyrus, retrosplenial cortex, and temporal fusiform cortex. These voxels largely fell within the medial temporal subsystem (50% of voxels within DMN) and core subsystem (40% of DMN voxels); there was less overlap with the dorsomedial DMN subsystem (10%). In contrast, greater task focus in reading was correlated with increased deactivation in overlapping regions in bilateral middle and medial frontal gyrus, frontal pole, insular cortex, anterior/posterior cingulate gyrus, precuneus and angular gyrus. These voxels largely fell within core DMN (82% of DMN voxels), with limited overlap in dorsomedial and medial temporal DMN subsystems (13% and 5% of DMN voxels, respectively). Therefore, when people are more focussed on autobiographical memories, there is increased activation in core and medial temporal DMN regions, while deactivation of core DMN regions is linked to greater mental focus during reading. Taken together, this analysis establishes that regions of ventromedial prefrontal cortex, posterior cingulate cortex, and superior frontal gyrus contribute to better focus on autobiographical memory retrieval while compromising the ability to read for comprehension.

## Experiment 2

Experiment one established that our paradigm captured the expected mutual inhibition between reading and autobiographical memory retrieval (*Figure 1*) and found that both states depend on activity within distinct regions within the DMN (*Figures 2 and 3*). Using two resting-state datasets, the aim of Experiment two was to understand (i) whether these DMN regions associated with reading and autobiographical memory retrieval show differences in their intrinsic functional connectivity to ventral

visual regions important for reading; and (ii) how individual differences in the connectivity of these DMN seeds relate to the tendency to mind-wander during reading in a more naturalistic setting. In order to directly test the similarity in connectivity across task and rest states, Experiment 2 took the peak DMN activations for reading and autobiographical memory retrieval obtained from Experiment 1 as seed locations. In Dataset 1, we considered intrinsic connectivity differences between these DMN seeds across the whole-brain and within reading-related visual cortex identified in Experiment 1. Moreover, as our whole-brain analysis identified a site in visual cortex that was more decoupled in people who reported more contents of mind-wandering in Dataset 2, we also performed an additional connectivity analysis of the task data from Experiment 1, to establish the extent to which these connectivity patterns at rest, found to be related to individual variance in naturalistic mind-wandering during reading in Experiment 2, also occur during task-induced autobiographical memory retrieval.

## Results of resting-state intrinsic connectivity

Our first analysis examined the extent to which DMN subnetworks linked to different task states are functionally connected to regions of the ventral visual stream activated during reading. This analysis helps us to determine whether the reductions in ventral visual cortex seen in Experiment 1 during autobiographical memory retrieval solely reflect an attentional phenomenon that emerges because participants were asked to attend to memory retrieval rather than textual input, or whether these effects relate to differences in the intrinsic functional architecture of the DMN regions important for autobiographical memory retrieval and reading. To address this question, we conducted whole-brain resting-state functional connectivity analyses targeting the peak activation regions of the DMN, since they represent regions with the most differential activation associated with reading and autobiographical memory retrieval respectively (see Materials and methods). We created a reading peak seed that fell within dorsomedial DMN (MNI coordinates: −56, −10, −12) and an autobiographical memory retrieval peak seed (MNI coordinates: −6, −50, 22) that was within core DMN, by placing a binarised spherical mask with a radius of 3 mm, centred on the MNI coordinates in these selected sites. The results of this analysis can be seen in *Figure 4A*; regions more strongly correlated with the DMN seed linked to reading are presented in warm colours, while those regions showing greater functional connectivity with the DMN seed linked to autobiographical memory retrieval are shown in cool colours. The reading DMN region showed greater intrinsic connectivity to regions of ventral visual cortex. We also performed a follow-up ROI analysis to examine whether there was differential intrinsic connectivity to a visual region activated during reading in Experiment 1 (MNI coordinates: −32, −92, −10), comparing DMN seeds located at peak reading and autobiographical memory retrieval sites. A paired samples *t*-test revealed stronger intrinsic connectivity for the reading than the autobiographical memory DMN seed to this visual site implicated during reading (Reading DMN-to-Reading visual: Mean ± *SD* = 0.02 ± 0.14, Autobiographical memory DMN-to-Reading visual: Mean ± *SD* = −0.01 ± 0.14; *t*(242) = 2.78, *p* = 0.006). This analysis provides further evidence that the DMN and visual regions showing stronger activation during reading than memory retrieval also had stronger intrinsic connectivity at rest. Moreover, we conducted a formal conjunction analysis between the regions showing stronger intrinsic connectivity with DMN seeds linked to reading versus autobiographical memory retrieval and the spatial map of stronger activity during reading versus autobiographical memory retrieval. The results of this analysis are presented in *Figure 4B*. This 'task-rest' conjunction establishes that the functional connectivity of regions linked to reading is mirrored by their joint activation during reading, suggesting that co-activation of aspects of DMN and visual cortex during reading is at least partially rooted in their intrinsic functional organisation. A supplementary analysis also revealed patterns of structural connectivity supporting this connection between ventral visual regions and areas of DMN that support reading (see *Appendix 1—figure 8*).

Our second analysis examined how the functional architecture of these DMN seeds associated with reading and autobiographical memory retrieval was related to individual variation in naturally occurring mind-wandering during reading as measured in our prior study (*Zhang et al., 2019*). In that study, we demonstrated that DMN-to-visual decoupling is associated with individual differences in mind-wandering experiences while reading, with greater decoupling of middle temporal DMN from medial visual cortex for individuals with more frequent off-task thoughts (*Zhang et al., 2019*). In the current analysis, we hoped to ascertain the similarities between the experimental situation in Experiment 1 (which mimicked processes linked to mind-wandering) and this more naturalistic context.

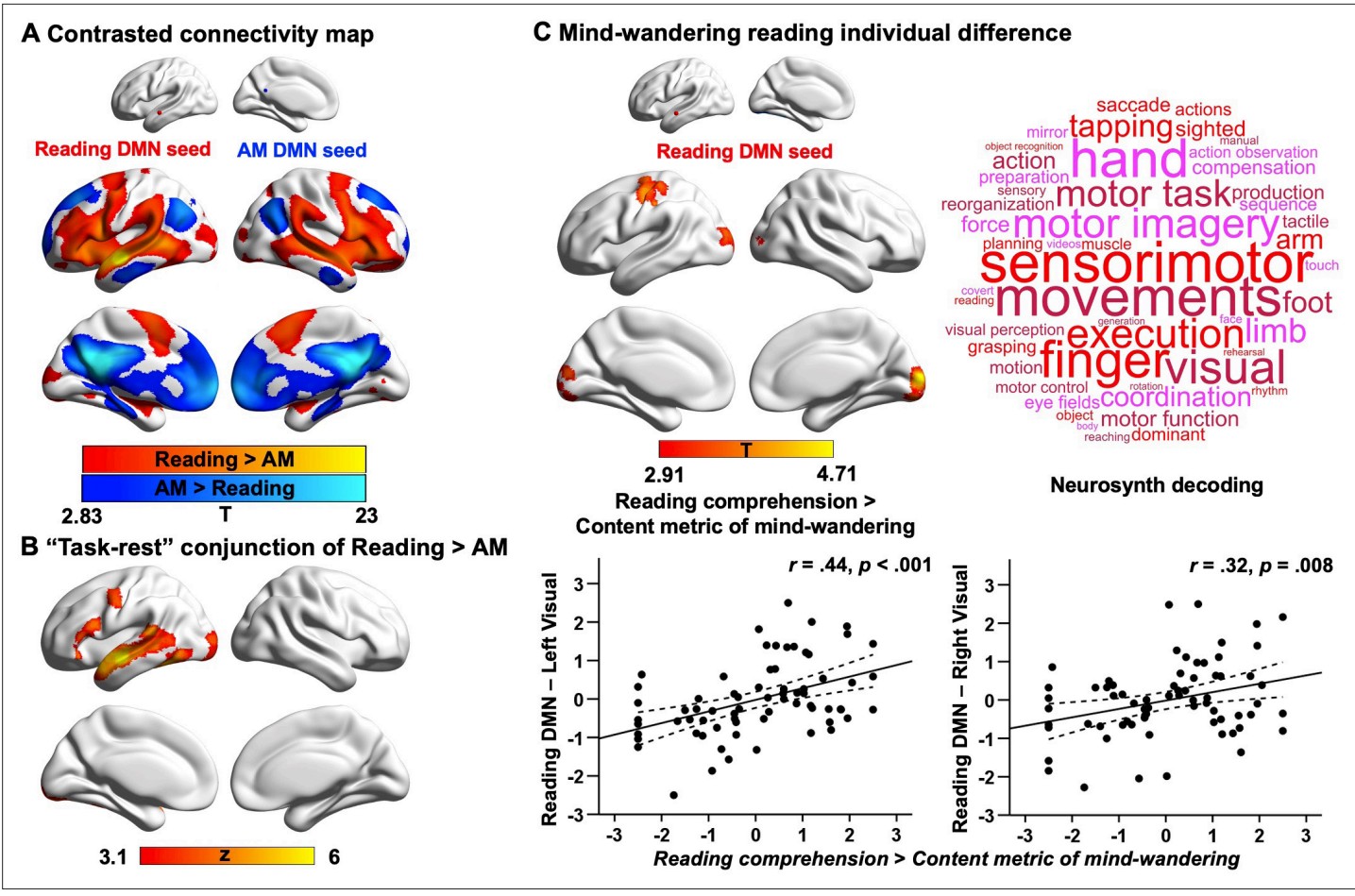

**Figure 4.** Results of resting-state intrinsic connectivity. Analyses examining the functional architecture of DMN regions associated with reading and autobiographical memory (AM) retrieval and their relation to individual differences in naturally occurring mind-wandering during reading. Panel (**A**) shows the results of a functional connectivity analysis examining differences between the DMN seeds (reading DMN MNI coordinates: − 56,–10, –12; autobiographical memory DMN MNI coordinates: − 6,50, 22). Regions showing stronger intrinsic connectivity to the reading DMN seed are shown in warmer colours, while regions showing stronger intrinsic connectivity to the autobiographical memory DMN seed are shown in cooler colours. The lower panel (**B**) shows results of a formal conjunction between regions associated with greater activity during reading versus autobiographical memory retrieval, and regions showing stronger correlation at rest with DMN seed activated by reading. Panel (**C**) shows the relationship of these seeds' functional architecture and self-reports of mind-wandering during reading. Group-level regression, using the DMN site showing peak activation during reading as a seed, demonstrated stronger connectivity with regions in primary visual cortex and postcentral gyrus in individuals with good comprehension and less reported mind-wandering content. To visualise this effect, the scatterplots present the correlation between behaviour and the correlation between the reading-relevant DMN seed and the identified visual clusters (beta values). The error lines on the scatterplots indicate the 95% confidence estimates of the mean. Each point describes an individual participant. The word cloud shows the functional associations with this connectivity map identified using Neurosynth. (The unthresholded functional connectivity maps contrasting reading and autobiographical memory DMN seeds from (**A**), the conjunction map in (**B**), and the mind-wandering individual difference map in (**C**), used for Neurosynth decoding, are provided in Neurovault at https://neurovault.org/collections/9432/).

We included individual differences in reading comprehension and off-task thoughts (i.e., frequency and content of mind-wandering including autobiographical memory; see *Appendix 1—figure 6*) as explanatory variables in a group-level regression and used the peak locations within the DMN that were linked to autobiographical memory retrieval and reading from Experiment 1 as seeds (with the reading seed in a different location within lateral temporal gyrus relative to our previous study). When seeding the reading DMN peak, the contrast of reading comprehension over mind-wandering content revealed regions of left primary visual cortex (corrected cluster-size $p$-FWE = 0.004) and right primary visual cortex (corrected cluster-size $p$-FWE = 0.004), as well as postcentral gyrus (corrected cluster-size $p$-FWE = .006), which showed stronger connectivity for individuals with better reading comprehension and less mind-wandering content reported in a questionnaire (see *Figure 4C*), in line with our previous

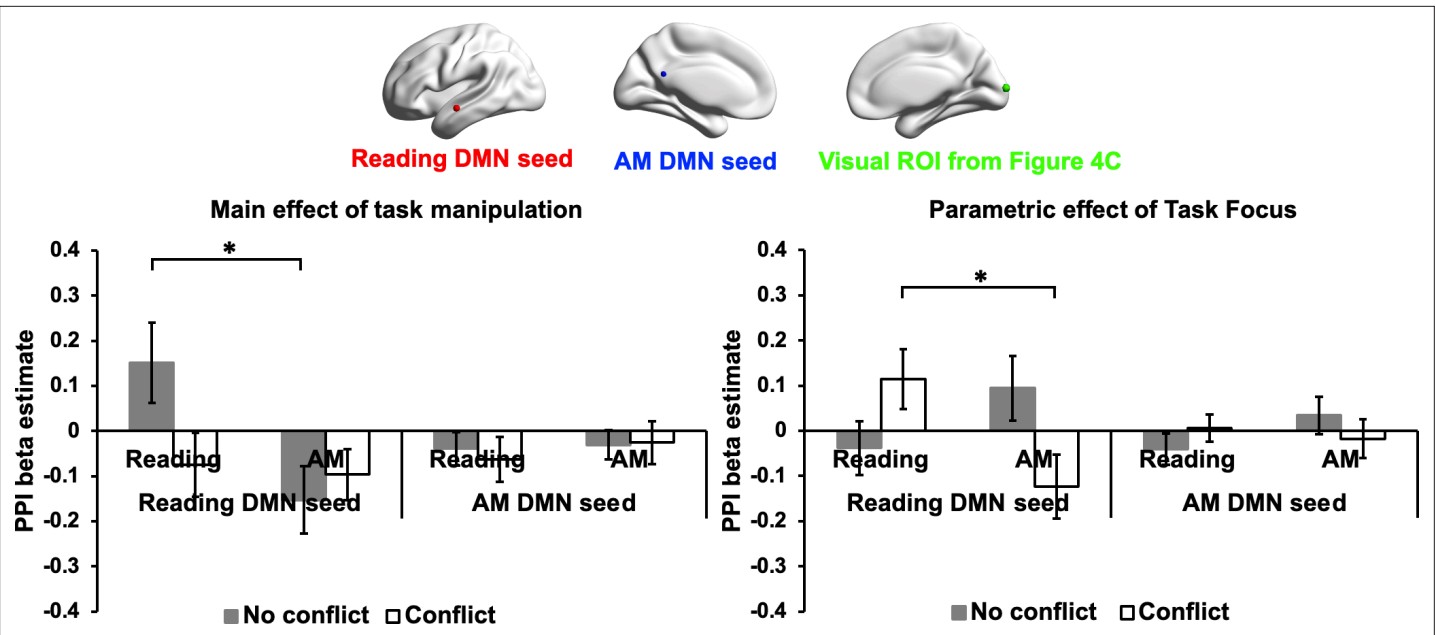

**Figure 5.** Results of task-based functional connectivity in visual ROI. These plots present the functional connectivity in Experiment 1 of reading and autobiographical memory DMN seeds with a visual ROI that showed more decoupling for people with a propensity for mind-wandering while reading in Experiment 2 Dataset 2 (MNI coordinates: 8, -98, 8; identified in the contrast of Reading comprehension > Mind-wandering content in *Figure 4C*). The bar charts plot the mean PPI $\beta$ values (i.e., representing the strength of functional connectivity between each DMN seed and the visual ROI). The left-hand graph shows the main effect of our task manipulation over the implicit baseline, while the right-hand graph shows the parametric effect of task focus for each experimental condition. Error bars depict the standard error of the mean. * indicates Bonferroni-corrected *p* value < 0.05.

findings (*Zhang et al., 2019*). To confirm the most likely functional associations with this connectivity map, we performed meta-analytic decoding using Neurosynth. The results of this analysis are presented in *Figure 4C* in the form of word clouds where the font size describes the strength of the relationship. The decoding of this connectivity map yielded terms largely linked to perception, such as 'sensorimotor' and 'visual'. In summary, weaker intrinsic connectivity between visually coupled DMN regions and the visual cortex is associated with a tendency to mind-wander while reading. Importantly, this analysis establishes a link between our experimental manipulation that mimics key processes engaged during mind-wandering in reading, and a metric describing a naturally occurring example of the state. No behavioural associations were recovered when using the DMN seed linked to autobiographical memory retrieval as a seed. We re-ran all these analyses using a more stringent motion criterion, excluding individuals with mean head motion that exceeded 0.2 mm. The same pattern of results was obtained. These results are presented in the Appendix 1.

## Results of task-based functional connectivity

The resting-state connectivity analysis revealed a relationship between individual variation in the tendency for mind-wandering during naturalistic reading and decoupling of reading-related DMN from visual cortex. To better understand whether this pattern of decoupling occurs in a similar way during experimental situations that mimic the experience of mind-wandering during reading, we performed psychophysiological interaction (PPI) analysis to examine the functional connectivity of the same regions in the task-based neuroimaging data recorded in Experiment 1. Two PPI models were conducted for each of the reading and autobiographical memory DMN seeds, examining connectivity during each experimental condition, as well as the parametric effect of task focus on connectivity (*Figure 5*; see Materials and methods for PPI analysis). An ROI approach was used to examine the estimated connectivity for each participant between the reading and autobiographical memory DMN seeds and an ROI in primary visual cortex that showed greater decoupling in participants with a tendency to mind-wander during reading in Experiment 2; we used a spherical ROI centred on the peak activation in the visual cluster in *Figure 4C* (MNI coordinates: 8,–98, 8; see *Figure 5*). For each model, a 2 (Task: Reading vs. Autobiographical memory) by 2 (Conflict: No conflict vs. Conflict)

repeated-measures Analysis of Variance (ANOVA) was performed to examine differences in functional connectivity across conditions.

For the Reading DMN seed, when considering the overall effects of task on connectivity, there were no significant main effects of Task, $F(1,27) = 3.51$, $p = 0.072$, $\eta_p^2 = .12$, or Conflict, $F(1,27) = 1.87$, $p = 0.18$, $\eta_p^2 = .07$. However, there was a significant interaction between Task and Conflict, $F(1,27) = 5.42$, $p = 0.028$, $\eta_p^2 = .17$. Post-hoc $t$-tests revealed a significant effect of Task in the absence of conflict, $t(27) = 2.81$, Bonferroni-corrected $p = 0.018$ for two tests, reflecting higher connectivity between the reading DMN seed and visual cortex when participants were reading. There was no effect of Task in the conflict condition, $t(27) = 0.21$, Bonferroni-corrected $p = 1$ for two tests, consistent with the view that conflict from autobiographical memory can disrupt visual connectivity to DMN during reading. *Note*: Twenty-eight participants were included in this analysis. One participant was excluded because the reading DMN seed location failed to generate a time-series for connectivity analysis. In addition, one outlying connectivity value for the reading DMN seed to visual cortex was identified in the Pure Reading condition and replaced with the mean of that condition. The same patterns were found when replacing this outlier and when using the original extreme value.

For the parametric effects of Task Focus, there were no significant main effects of either Task, $F(1,27) = 0.61$, $p = 0.44$, $\eta_p^2 = .02$, or Conflict, $F(1,27) = 0.19$, $p = 0.67$, $\eta_p^2 = .007$. There was a significant interaction effect between these two factors, $F(1,27) = 6.54$, $p = 0.016$, $\eta_p^2 = .20$. Post-hoc $t$-tests found a significant main effect of Task in the conflict condition, $t(27) = 2.62$, Bonferroni-corrected $p = 0.030$ for two tests, when participants were more focussed on reading in the face of distraction from autobiographical memory cues, connectivity between the reading DMN seed and visual cortex was higher. There was no difference between the tasks in the absence of conflict, $t(27) = -1.25$, Bonferroni-corrected $p = 0.44$ for two tests. No effects were found for the autobiographical memory DMN seed (All $F > 1.5$). These results are summarised in *Figure 5*.

Taken together, these results suggest that reading-related DMN decouples from visual cortex when participants retrieve personally relevant memories in the absence of sentences. When reading and memory retrieval are in conflict, these sites couple when participants are more focussed on reading and decouple when participants are more focussed on autobiographical memories. Importantly, this analysis establishes a similar perceptual decoupling pattern for this visual site in both naturally occurring and experimentally mimicked experiences of mind-wandering during reading, highlighting the importance of perceptual decoupling when memory retrieval competes with reading.

## Discussion

Our study set out to understand how the experience of mind-wandering while reading creates a state of perceptually decoupled thought that derails our comprehension of the text. In particular, we tested an emerging hypothesis that spans both psychological and neural domains. At the psychological level, contemporary theories of self-generated states suggest that functional decoupling from perceptual input is important for effective memory retrieval, providing a process account of why mind-wandering during reading can derail comprehension (*Smallwood, 2013*). In the neural domain, we draw on a recent hypothesis that the role of DMN subnetworks in human cognition is related to their topographic location on the cortical mantle (*Margulies et al., 2016*; *Smallwood et al., 2021a*). These regions occupy locations that are both the terminus of processing streams within the cortex important for abstract forms of cognition such as reading, and yet at a distance along the cortical surface from input systems, explaining why they can also be engaged by situations in which mental content is broadly unrelated to perceptual input (*Smallwood et al., 2021a*).

Experiment 1 established a pattern of mutual inhibition between the act of reading for comprehension and the concurrent retrieval of autobiographically-relevant content. This pattern of mutual inhibition parallels the well-established negative correlation between naturally occurring mind-wandering and an individual's ability to comprehend what they are reading (*McVay and Kane, 2012*; *Smallwood et al., 2008*). Using fMRI to index brain activity, we identified that these two states differentially recruited different aspects of the DMN in direct contrasts of these two tasks, with greater recruitment of the dorsomedial DMN subnetwork when participants were reading for comprehension, and greater activity within the core of the DMN during autobiographical memory retrieval (See also *Chiou et al., 2020*). The involvement of the DMN in both self-generated mental content and reading for comprehension is consistent with prior studies exploring trait variation in mind-wandering while reading

(*Smallwood et al., 2013*; *Zhang et al., 2019*). Importantly, Experiment one found that ventral visual regions were engaged when participants were reading for comprehension as opposed to engaged in autobiographical memory recall, a pattern consistent with the view that the retrieval of autobiographical memories during mind-wandering creates a perceptually decoupled state (*Schooler et al., 2011*; *Smallwood, 2013*).

Experiment 2 used resting-state functional connectivity to establish that there is strong intrinsic coupling between regions of DMN relevant to reading and ventral visual cortex; in contrast, regions of core DMN, activated by autobiographical memory retrieval, showed reduced correlation with these ventral visual regions, compared with aspects of DMN linked to reading. This pattern suggests that the reduction of activity in ventral visual cortex observed in Experiment 1 was not an artefact of our task instructions which required participants to attend to autobiographical memories rather than textual input. Instead, the core DMN regions activated when we retrieve autobiographical memories are functionally distant from regions of ventral visual cortex important for reading comprehension. Moreover, individual differences in intrinsic connectivity associated with the state of mind-wandering during reading are consistent with this view. We found that, in individuals who remained focussed on reading and who had better comprehension, primary visual cortex showed stronger functional coupling to the aspects of DMN that support reading. For people who generated more off-task thought contents, these regions of visual cortex showed weaker functional coupling to reading-relevant DMN regions. This perceptual decoupling pattern was also seen in our task-based functional connectivity analysis of Experiment 1, since we found that decoupling of a reading-related DMN site to visual cortex was greater during the retrieval of autobiographical memories in the absence of sentences and when there was better focus on memory retrieval in the face of distracting semantic inputs. These converging lines of evidence demonstrate the importance of perceptual decoupling for mind-wandering during reading, which consequently disrupts the pattern of perceptual coupling that supports reading comprehension (*Smallwood, 2011*).

Our analyses have important implications for psychological theories regarding self-generated states such as mind-wandering, as well as for our understanding of the involvement of the DMN in human cognition. Psychologically, our study suggests that simply asking individuals to retrieve autobiographical information creates a perceptually decoupled state that is at odds with the comprehension of information from the external environment. This pattern is consistent with the view that perceptual decoupling is related to the persistence of self-generated mental content in consciousness (*Smallwood, 2013*). In the neural domain, our data suggests that mind-wandering while reading involves a shift in the balance of neural activity within the DMN, away from lateral temporal and prefrontal regions that are closely linked to regions of ventral visual cortex important for reading, and towards regions within the core of this system involved in supporting mental content that is unrelated to the external environment during autobiographical memory retrieval. In this context, it is worth noting that studies that have established the role of the DMN in explicit memory retrieval also show that this pattern is accompanied by the suppression of visual processing (*Murphy et al., 2019*). Furthermore, individuals with epilepsy who are impaired in the process of pattern separation necessary for accurate episodic memory retrieval show atypical suppression of brain activity in visual cortex during memory retrieval (*Li et al., 2021*). The current data, therefore, provide novel support for the possibility that antagonistic activity patterns in the DMN and in sensory cortices may be important for features of memory retrieval to proceed in an effective manner.

More generally, our data add to growing evidence for a broad contribution of the DMN to features of human cognition. In particular, our findings are consistent with observations that the broader DMN can support apparently antagonistic states, particularly both perceptually coupled and decoupled modes of cognition. Initial focus on the DMN assumed that this system was primarily important for internally focused states. Recently, however, *Yeshurun et al., 2021* have argued that the DMN plays a key role in the integration of internal and external information in the service of aligning the perspectives of different individuals over time. Our data highlighting the role of the broader DMN in reading for comprehension is consistent with this perspective, as this mode of operation could help to create common ground between different individuals in their understanding of a narrative. However, the involvement of the same broad system in situations such as mind-wandering suggests that it can also lead to a breakdown in the common ground between individuals, in this case by impairing comprehension during reading. Both of these operational modes stem from topographic location of DMN at

the end of sensory processing streams, such as the ventral visual stream. This allows this network to provide abstract representations of external information and, at the same time, the DMN's functional and structural distance to unimodal sensory systems provides the opportunity for decoupled states to emerge, which support mental contents that are unrelated to external input (such as self-generated off-task states; *Margulies et al., 2016*; *Smallwood et al., 2021a*).

Although different aspects of the DMN play distinctive roles in reading comprehension and autobiographical memory retrieval, there are still some neural similarities between these two mental states; since the lateral temporal and frontal regions within dorsomedial DMN activate in response to both tasks (i.e., identified by a formal conjunction across the contrast maps of each task over the letter string baseline; see *Appendix 1—figure 4*). Previous studies have shown the engagement of lateral temporal and inferior frontal cortex in both reading and memory recall (*Andrews-Hanna et al., 2014a*; *Ferstl et al., 2008*; *Lambon Ralph et al., 2017*; *Summerfield et al., 2009*; *Svoboda et al., 2006*; *Yang et al., 2019*). Given that both tasks involve semantic cognition (e.g. *Dehaene et al., 2015*; *Graham et al., 2003*; *Spitsyna et al., 2006*; *Svoboda et al., 2006*), this finding is consistent with prior work implicating lateral temporal cortex and inferior frontal gyrus in the representation and retrieval of heteromodal conceptual knowledge (*Badre et al., 2005*; *Binder et al., 2009*; *Jefferies, 2013*; *Lambon Ralph et al., 2017*; *Noonan et al., 2013*). In addition, we found that the regions important for reading comprehension activate not only when visual input is task-relevant (i.e., during reading for comprehension), but also when this input is irrelevant to the ongoing task (i.e., revealed by *autobiographical memory retrieval with conflict > autobiographical memory retrieval with no conflict*; see *Appendix 1—figure 5*), and irrespective of task focus. In line with our findings, these reading-relevant regions may be more perceptually coupled supporting visual to conceptual knowledge mapping. In this way, they might be sensitive to situations in which meaning emerges in the external environment, even when the focus of attention is elsewhere.

Although our study provides important insight into how the occurrence of autobiographical mental content can derail our ability to make sense of the external environment, it also leaves open several important questions. First, in both Experiment 1, and in many examples of naturally occurring mind-wandering, the off-task mental contents may have greater relevance to the individual than information in the environment (*Sanders et al., 2017*; *Smallwood et al., 2011*), perhaps because these states rely on ventral regions of medial prefrontal cortex (*Konu et al., 2020*) that are important for motivated states (*Kouneiher et al., 2009*; *Rushworth et al., 2004*; *Stawarczyk and D'Argembeau, 2015*). It is therefore unclear whether reading more engaging text would change the likelihood of off-task thoughts emerging, and the neural systems that are engaged during reading. To address this issue, it would be useful for future studies to explore the neural systems recruited by highly engaging and personally-relevant texts, to establish if these are more similar to those observed when reflecting on autobiographical memories. Second, differing from our current design, one's naturalistic reading is self-paced and involves longer passages with a complex pattern of attentional focus on the text. People sometimes slow down or speed up their reading, as attention, interest or complexity wax and wane, and often, they re-read earlier parts of text. Although there are important discrepancies between the mind-wandering occurring during naturalistic reading and the design of Experiment 1, our study established important neural similarities between task-induced autobiographical memory (Experiment 1) and naturally occurring mind-wandering (Experiment 2) during reading. These similarities cannot be accounted for by differences between the experimental situations and therefore our study establishes important support for the process-occurrence view of self-generated states (*Smallwood, 2013*). Third, contemporary studies suggest that different types of self-generated thoughts have different neural correlates (*Karapanagiotidis et al., 2020*; *Mckeown et al., 2020*). Accordingly, it is possible that certain features of naturally occurring off-task thought patterns could lead to greater or lesser disengagement from external input. This question could be readily addressed by combining multi-dimensional experience sampling (*Smallwood et al., 2016*) with brain activity recorded while individuals read.

## Materials and methods

### Participants

A total of 339 participants were recruited in this study. For Experiment 1, 29 undergraduate students were recruited (age-range 18–23, mean age ± SD = 20.14 ± 1.26, 6 males). For Experiment 2, we used two separate resting-state samples: one sample consisted of 244 participants (age-range 18–31, mean age ± SD = 20.73 ± 2.39, 77 males; three participants overlapped with Experiment 1), which was used to examine the intrinsic connectivity of DMN regions linked to reading and autobiographical memory retrieval. One participant was excluded from this analysis due to excessive head motion (i.e., mean head motion > 0.4 mm). Another published dataset of 69 participants (age range 18–31, mean age ± SD = 19.87 ± 2.33, 26 males; without any participant overlap with the other two samples) was used to characterise the relationship of functional architecture of the DMN regions associated with reading and autobiographical memory retrieval and individual variation in naturally occurring mind-wandering reading, as measured in our prior study (*Zhang et al., 2019*). All were right-handed native English speakers, and had normal or corrected-to-normal vision. None had any history of neurological impairment, diagnosis of learning difficulty or psychiatric illness. All provided written informed consent prior to taking part and received monetary/course credits compensation for their time. Ethical approval was obtained from the Research Ethics Committees of the Department of Psychology and York Neuroimaging Centre, University of York (Project number: P1406).

### Materials

144 highly imageable, frequent and concrete nouns were selected to serve as key words within sentences and as cue words for autobiographical memory recall. These nouns were divided into two lists (i.e., 72 words for each task) that did not differ in terms of frequency (CELEX database; *Baayen et al., 1993*), imageability (*Davis, 2005*), and concreteness (*Brysbaert et al., 2014*; *p* > 0.1). The sentences were constructed by using these key words as a search term in Wikipedia to identify text that described largely unfamiliar facts about each item; in this way, the contents of these sentences were neutral in valence (Sentence Length: Mean ± *SD* = 20.04 ± 0.93 words). For example, "*Posters are also used for reproductions of artwork, particularly famous works, and are low-cost compared to the original artwork*" for the keyword POSTER, *and "All mammals have some hair on their skin, even marine mammals like whales and dolphins, which appear to be hairless*" for the keyword MAMMAL. These sentences and the autobiographical memory cues were then divided into three sets and assigned to different conditions (with this assignment counterbalanced across participants). The sentences were assigned to (1) *Reading* without conflict from memory recall; (2) *Reading* with conflict from memory recall, and (3) *Memory Recall* with conflict from concurrent sentence presentation. Similarly, the autobiographical memory cues were assigned to (1) *Memory Recall* without conflict from sentences and (2) *Memory Recall* with conflict from sentences, as well as (3) *Reading* with conflict from memory recall. In addition, the key words used in these conditions were matched on key psycholinguistic variables: they did not differ in lexical frequency, imageability, or concreteness (all *F* < 1.07). In addition, all the words in the three sets of sentences were comparable across these variables (see *Appendix 1—table 1*; all *F* < 1.40). Two additional cue words were created for task practice.

### Task procedure for reading and autobiographical memory retrieval task in Experiment 1

Testing occurred across two consecutive days. On Day 1, participants were asked to generate their own personal memories from cue words (e.g., Party) outside the scanner. They were asked to identify specific events that they were personally involved in and to provide as much detail about these events as they could, including when and where the event took place, who was involved in, what happened, and the duration. They were asked to type these details into a spreadsheet, which ensured that comparable information was recorded for different cue words.

On the following day, participants were asked to read sentences for comprehension, or to recall their generated personal memories inside the scanner. In reading trials, sentences were presented word by word, after either (1) an autobiographical memory cue word (e.g., Party), creating conflict between reading and personal memory retrieval, or (2) a letter string (e.g., XXX) allowing reading to take place in the absence of conflict from autobiographical memory. We controlled the duration of

the sentences by presenting the words on 15 successive slides, combining short words on a single slide (e.g., *have been* or *far better*) or presenting articles and conjunctions together with nouns (e.g., *the need; and toys*). In memory recall trials, participants were asked to recall autobiographical memories during the presentation of either (1) an unrelated sentence, creating conflict from task-irrelevant semantic input, or (2) letter strings (e.g., XXX) allowing autobiographical memory to take place without distracting semantic input. As a control condition, meaningless letter strings (e.g., xxxxx) were presented. In order to ensure the participants were maintaining attention to the presented stimuli (even when these were irrelevant and creating competition), they were told to press a button when they noticed the colour of a word or letter string change to red. There were 3 trials out of 24 trials in each condition that involved responding in this way. Behavioural data for the colour change detection task are presented in *Appendix 1—figure 1*.

As shown in *Figure 1*, each trial started with a fixation cross (1–3 s) in the centre of the screen. Then either an autobiographical memory cue word or a letter string appeared for 2 s. During the presentation of the cue word, participants were asked to bring to mind their personal memories related to this item. Next, the task instruction (i.e., READING or MEMORY RECALL) was presented for 1 s. Following that, words from sentences or letter strings were presented, with each one lasting 600ms. On memory recall trials, participants were asked to keep thinking about their autobiographical memory, as much detail as possible, until the end of the trial.

After each trial, participants were asked to rate several dimensions of their experience. In the reading task, they were asked about task focus, their comprehension, and conceptual familiarity. For autobiographical memory retrieval trials, they were asked about task focus, vividness, and how consistent their retrieval was to the memory they specified day before. The three rating questions were sequentially presented after a jittered fixation interval lasting 1–3 s. Participants were required to rate these characteristics on a scale of 1 (not at all) to 7 (very well) within 4 s for each question. There were no ratings for the letter string trials.

Stimuli were presented in four runs, with each containing 30 trials: six trials in each of the four experimental conditions, and six letter string trials. Each run lasted 12.85 min, and each reading or memory recall trial lasted in the range of 26.2–29.9 s with an average of 28.0 s, while each letter string trial lasted from 13.0 to 15.0 s with an average of 14.0 s. In addition, trials were presented in a pseudorandom order to ensure trials from the same experimental condition were not consecutively presented more than three times.

Before entering the scanner, participants completed a 6-min task to test their memory of the personal events they generated the day before scanning. They were also asked to review their generated memories and refresh themselves with the ones that were not well remembered. Next, they completed an eight-trial practice block containing all types of conditions to ensure fully understanding of the task requirements.

## Behavioural assessment of mind-wandering reading in Experiment 2

This dataset was used in our previous study (*Zhang et al., 2019*). Participants were asked to complete a battery of behavioural assessments examining their reading comprehension and off-task thought, while they read a passage about the topic of geology. During reading, they were required to note down any moments when they noticed they had stopped paying attention to the meaning of the text. After they finished reading, they were asked to answer 17 open-ended questions to assess their comprehension of the text, without being able to refer back to the text. A self-reported measurement, with 22 questions about the content of thoughts (e.g., *I thought about personal worries*), was used to assess off-task behaviour during the reading task (see *Appendix 1—figure 6*). This analysis revealed that people were thinking about autobiographical memories (past events) alongside future events, other people and emotions, when they reported mind-wandering during reading. In this way, both off-task thoughts (i.e., frequency and the content of these experiences) and reading comprehension were assessed.

## Neuroimaging data acquisition

Structural and functional data were acquired using a 3T GE HDx Excite MRI scanner utilizing an eight-channel phased array head coil. Structural MRI acquisition in all participants was based on a T1-weighted 3D fast spoiled gradient echo sequence (repetition time (TR) = 7.8 s, echo time (TE) =

minimum full, flip angle = 20°, matrix size = 256 × 256, 176 slices, voxel size = 1.13 × 1.13 × 1 mm³).
The task-based activity was recorded using single-shot 2D gradient-echo-planar imaging sequence
with TR = 3 s, TE = minimum full, flip angle = 90°, matrix size = 64 × 64, 60 slices, and voxel size = 3 ×
3 × 3 mm³. In Experiment 1, the task was presented across four functional runs, with each containing
257 volumes. In Experiment 2, using the same scanning parameters, a 9-min resting-state fMRI scan
was recorded, containing 180 volumes. The participants were instructed to focus on a fixation cross
with their eyes open and to keep as still as possible, without thinking about anything in particular.

## Pre-processing of task-based fMRI data in Experiment 1

All functional and structural data were pre-processed using a standard pipeline and analysed via the
FMRIB Software Library (FSL version 6.0, https://www.fmrib.ox.ac.uk/fsl; RRID: SCR_002823). Individual T1-weighted structural brain images were extracted using FSL's Brain Extraction Tool (BET).
Structural images were linearly registered to the MNI152 template using FMRIB's Linear Image
Registration Tool (FLIRT). The first three dummy volumes of each functional scan were removed in
order to minimise the effects of magnetic saturation. The functional neuroimaging data were analysed using FSL's FMRI Expert Analysis Tool (FEAT). We applied motion correction using MCFLIRT
(*Jenkinson et al., 2002*), slice-timing correction using Fourier space time-series phase-shifting
(interleaved), spatial smoothing using a Gaussian kernel of FWHM 6 mm, and high-pass temporal
filtering (sigma = 100 s) to remove temporal signal drift. In addition, motion scrubbing (using the
fsl_motion_outliers tool) was applied to exclude volumes that exceeded a framewise displacement
threshold of 0.9 mm.

## Pre-processing of resting-state fMRI data in Experiment 2

Pre-processing was performed using the CONN-fMRI functional connectivity toolbox (RRID:
SCR_009550), Version 18a (http://www.nitrc.org/projects/conn; *Whitfield-Gabrieli and Nieto-Castanon, 2012*), based on Statistical Parametric Mapping 12 (http://www.fil.ion.ucl.ac.uk/spm/).
Participants' motion estimation and correction were then carried out through functional realignment
and unwarping, and potential outlier scans were identified using the Artifact Detection Tool (ART)
toolbox (https://www.nitrc.org/projects/artifact_detect). Structural images were segmented into
Grey matter, White matter and Cerebrospinal Fluid tissues and normalized to the MNI space with
the unified segmentation and normalization procedure (*Ashburner and Friston, 2005*). Functional
volumes were slice-time (bottom-up, interleaved) and motion-corrected, skull-stripped and co-registered to the high-resolution structural image, spatially normalised to MNI space using the unified-segmentation algorithm (*Ashburner and Friston, 2005*), smoothed with an 8 mm FWHM Gaussian
kernel.

Pre-processing steps automatically created three first-level covariates: a *realignment* covariate
containing the six rigid-body parameters characterising the estimated subject motion for each participant, a *scrubbing* covariate containing the potential outliers scans for each participant (i.e., identified
through the artefact detection algorithm included in CONN, with intermediate settings: scans for each
participant were flagged as outliers based on scan-by-scan change in global signal above z = 5, subject
motion threshold above 0.9 mm, differential motion and composite motion exceeding 97% percentile
in the normative sample), and a covariate containing quality assurance (QA) parameters (e.g., the
global signal change from one scan to another and the framewise displacement) for each participant.
Realignment parameters, potential outlier scans, signal from white matter and cerebrospinal fluid
masks and effect of rest (i.e., an automatically estimated trend representing potential ramping effects
in the BOLD timeseries at the beginning of the sessions), were entered as potential confound regressors into the model in the denoising step of the CONN toolbox. Using the implemented anatomical
CompCor approach (*Behzadi et al., 2007*), all these effects were removed within a single general
linear regression step to increase the signal to noise ratio in the functional images. Functional images
were then band-passed filtered (0.008–0.09 Hz) to constrain analyses to low-frequency fluctuations.
A linear detrending term was also applied, eliminating the need for global signal normalisation (*Chai
et al., 2012*; *Murphy et al., 2009*). Global signal regression was not performed because CompCor
can account for subject movement effects and other sources of noise in the BOLD signal (*Behzadi
et al., 2007*; *Muschelli et al., 2014*).

## Analysis of task-based fMRI data in Experiment 1

The pre-processed time-series data were modelled using a general linear model, using FMRIB's Improved Linear Model (FILM) correcting for local autocorrelation (*Woolrich et al., 2001*). Nine Explanatory Variables (EV) of interest and nine of no interest were modelled using a double-Gaussian hemodynamic response gamma function. The nine EVs of interest were: *Reading* (1) without and (2) with conflict from memory recall, *Autobiographical memory retrieval* (3) with and (4) without conflict from semantic input, (5) Letter String Baseline, (6-9) Task Focus effect for each of the four experimental conditions as a parametric regressor. Our EVs of no interest were: (10) Memory cue words and (11) Letter strings before the presentation of task instructions, Task instructions for *Reading* (12) without and (13) with conflict (this separation of the reading task instruction was based on the consideration that some recall or task preparation was likely to be occurring due to the presentation of autobiographical memory cues), plus task instructions for (14) *Memory Recall* and (15) *Letter String* baseline conditions. Other EVs of no interest were: (16) Fixation (the inter-stimulus fixations between the sentences or letter strings and the ratings questions), (17) Responses to catch trials (which included all time points with responses across conditions), and (18) Rating decision periods (including all the ratings across experimental conditions). EVs for each condition commenced at the onset of the first word of the sentence or the first letter string, with EV duration set as the presentation time (9 s). The parametric EVs for the effect of Task Focus during the target had the same onset time and duration as the EVs corresponding to the four experimental trials, but in addition included the demeaned Task Focus ratings value as a weight. The fixation period between the trials provided the implicit baseline.

We examined the main effects of Task, and Conflict for both the main experimental conditions and the effect of Task Focus, and comparisons of each experimental condition with the letter string baseline, which allowed us to identify the activation and deactivation in each task. We report the results of each condition over letter baseline, the activation and deactivation of each task, and effects of task conflict in *Appendix 1—figures 2 and 3*, and *Appendix 1—figure 5*. The four sequential runs were combined using fixed-effects analyses for each participant. In the group-level analysis, the combined contrasts were analysed using FMRIB's Local Analysis of Mixed Effects (FLAME1), with automatic outlier de-weighting (*Woolrich, 2008*). A 50% probabilistic grey-matter mask was applied. Clusters were thresholded using Gaussian random-field theory, with a cluster-forming threshold of $z = 3.1$ and a familywise-error-corrected significance level of $p < .05$.

## Analysis of resting-state fMRI data in Experiment 2

The functional connectivity analysis was performed using DMN peak seeds associated with reading and autobiographical memory retrieval. In a first-level analysis, we computed whole-brain seed-to-voxel correlations for each seed after the BOLD timeseries were pre-processed and denoised. For the group-level analysis of the dataset with 243 participants, we performed contrast between functional connectivity maps seeding from these two DMN seeds. For the group-level analysis of 69 participants, the EVs were entered into a GLM analysis, including reading comprehension scores, self-reported mind-wandering frequency, and the scores of the content of off-task thoughts (for details see *Zhang et al., 2019*). We examined both the main effects and contrasted effects of these behavioural measures. Group-level analyses in CONN were cluster-size FWE corrected at $p < 0.05$ (two-sided tests), and used a height threshold of $p < 0.005$. Bonferroni correction was also applied to account for the fact that we included two models, the cluster-size FWE $p$-value consequently accepted as significant was $p < 0.025$. Prior to data analysis, all behavioural variables were z-transformed and outliers more than 2.5 standard deviations above or below the mean were imputed with the cut-off value. All brain figures were created using BrainNet Viewer (http://www.nitrc.org/projects/bnv/; *Xia et al., 2013*).

## Psychophysiological interaction (PPI) analysis in Experiment 2

In order to understand whether the perceptual decoupling pattern at rest related to individual differences in mind-wandering identified in Experiment 2 also emerges in experimentally mimicked mind-wandering during reading, we conducted PPI analysis. Reading and autobiographical memory DMN seeds were created based on peak activation in each task and their time series were extracted after the BOLD timeseries were pre-processed. We then ran two separate models for each of these DMN seeds, which examined the main effect of the experimental condition (i.e., *Reading with no conflict, Reading with conflict, Autobiographical memory retrieval with no conflict,* and *Autobiographical*

*memory retrieval with conflict*), and the parametric effect of rated task focus in each condition on connectivity. These models included all the regressors in the basic task model of Experiment one described above (18 regressors and motion regressor), a PPI term for each of the four experimental conditions (based on the main effect of condition or the effect of task focus), as well as the time series of the DMN seeds, using the generalized psychophysiological interaction (gPPI) approach (*McLaren et al., 2012*). The regressors were not orthogonalized. The fixation period between the trials provided the implicit baseline. The four sequential runs were combined using fixed-effects analyses for each participant, which allowed us to extract the connectivity parameters for each experimental condition for each participant in each seed model.

## Neurosynth decoding

Task activation and conjunction maps were uploaded to Neurovault (https://neurovault.org/collections/9432/; *Gorgolewski et al., 2015*) and decoded using Neurosynth (*Yarkoni et al., 2011*). Neurosynth (RRID: SCR_006798) is an automated meta-analysis tool that uses text-mining approaches to extract terms from neuroimaging articles that typically co-occur with specific peak coordinates of activation. It can be used to generate a set of terms frequently associated with a spatial map (as in *Figure 2*). The results of cognitive decoding were rendered as word clouds using free online word cloud generator (https://www.wordclouds.com/). We manually excluded terms referring to neuroanatomy (e.g., 'inferior' or 'sulcus'), and repeated terms (e.g., 'semantic' and 'semantics').

## Acknowledgements

Funding: This work was supported by the European Research Council (Project ID: 771863 – FLEXSEM to EJ), and a China Scholarship Council (CSC) Scholarship (No. 201704910952 to MZ).

## Additional information

### Funding

| Funder | Grant reference number | Author |
| --- | --- | --- |
| European Research Council | Project ID: 771863 - FLEXSEM | Elizabeth Jefferies |
| China Scholarship Council | CSC Scholarship No. 201704910952 | Meichao Zhang |

The funders had no role in study design, data collection and interpretation, or the decision to submit the work for publication.

### Author contributions

Meichao Zhang, Conceptualization, Data curation, Formal analysis, Investigation, Methodology, Validation, Visualization, Writing – original draft, Writing – review and editing; Boris C Bernhardt, Formal analysis, Methodology; Xiuyi Wang, Dominika Varga, Katya Krieger-Redwood, Investigation; Jessica Royer, Raúl Rodríguez-Cruces, Reinder Vos de Wael, Formal analysis; Daniel S Margulies, Writing – review and editing; Jonathan Smallwood, Conceptualization, Supervision, Writing – original draft, Writing – review and editing; Elizabeth Jefferies, Conceptualization, Funding acquisition, Supervision, Writing – original draft, Writing – review and editing

### Author ORCIDs

Meichao Zhang http://orcid.org/0000-0001-9594-7229
Boris C Bernhardt http://orcid.org/0000-0001-9256-6041

### Ethics

Human subjects: Ethical approval was obtained from the Research Ethics Committees of the Department of Psychology and York Neuroimaging Centre, University of York (Project number: P1406). All participants provided written informed consent prior to taking part and received monetary/course credits compensation for their time.

Decision letter and Author response
Decision letter https://doi.org/10.7554/eLife.74011.sa1
Author response https://doi.org/10.7554/eLife.74011.sa2

## Additional files

### Supplementary files
• Transparent reporting form

### Data availability

Neuroimaging data at the group-level are openly available in Neurovault at https://neurovault.org/collections/9432/. Semantic material and script for the task are accessible in the Open Science Framework at https://osf.io/yvks7. Figure 2—source data 1 contains the numerical data used to generate the figures. The conditions of our ethical approval do not permit public archiving of the data because participants did not provide sufficient consent. Researchers who wish to access the data should contact the Research Ethics and Governance Committee of the York Neuroimaging Centre, University of York, or the corresponding authors. Data will be released to researchers when this is possible under the terms of the GDPR (General Data Protection Regulation).

The following dataset was generated:

| Author(s) | Year | Dataset title | Dataset URL | Database and Identifier |
|---|---|---|---|---|
| Zhang M | 2017 | Semantic material and script for the task are accessible | https://osf.io/yvks7/ | Open Science Framework, yvks7 |

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

# Appendix 1

## Supporting materials for Experiment 1
Linguistic properties of materials

**Appendix 1—table 1.** Linguistic properties of each set of key words within sentences and autobiographical memory cues, and the words within each set of sentences (Mean ± SD). Sets (i), (ii), and (iii) were counterbalanced across participants (see Materials).

| Conditions | Frequency | Imageability | Concreteness |
| --- | --- | --- | --- |
| (i) sentence key words | 1.31 ± .56 | 591.67 ± 34.20 | 4.74 ± .52 |
| (ii) sentence key words | 1.47 ± .50 | 598.05 ± 27.49 | 4.72 ± .55 |
| (iii) sentence key words | 1.29 ± .52 | 592.61 ± 44.35 | 4.64 ± .58 |
| (i) autobiographical memory cues | 1.59 ± .76 | 588.48 ± 41.20 | 4.70 ± .47 |
| (ii) autobiographical memory cues | 1.48 ± .62 | 594.64 ± 24.46 | 4.75 ± .30 |
| (iii) autobiographical memory cues | 1.54 ± .56 | 601.53 ± 23.65 | 4.73 ± .41 |
| (i) sentence materials | 2.59 ± .24 | 354.34 ± 27.41 | 2.72 ± .26 |
| (ii) sentence materials | 2.52 ± .20 | 352.40 ± 23.39 | 2.72 ± .21 |
| (iii) sentence materials | 2.48 ± .27 | 347.64 ± 35.24 | 2.80 ± .24 |

## Behavioural results of catch trials

Participants detected 75.6% of colour-change catch trials (i.e., they responded to this percentage of catch trials across conditions), showing that they were paying attention to inputs presented on the screen. Repeated-measures ANOVAs examining accuracy, response time (RT), and response efficiency (i.e., RT divided by accuracy), and assessing the effects of Task (Reading vs. Autobiographical memory recall) and Conflict (No conflict vs. Conflict), revealed that there were no differences in colour-change detection rates across conditions (see *Appendix 1—figure 1*); trials with no response were excluded from the RT analysis (24.4%). There was no main effect of Task (Accuracy: $F(1,28) = 1.54$, $p = 0.22$, $\eta_p^2 = .05$; RT: $F(1,28) = 1.92$, $p = 0.18$, $\eta_p^2 = .06$; Response efficiency: $F(1,28) = .35$, $p = 0.56$, $\eta_p^2 = .01$), no main effect of Conflict (Accuracy: $F(1,28) = 0.27$, $p = 0.61$, $\eta_p^2 = .01$; RT: $F(1,28) = 2.83$, $p = 0.10$, $\eta_p^2 = .001$; Response efficiency: $F(1,28) = 1.22$, $p = 0.28$, $\eta_p^2 = .04$), and no interaction (Accuracy: $F(1,28) = 0.85$, $p = 0.36$, $\eta_p^2 = .03$; RT: $F(1,28) = 1.27$, $p = 0.27$, $\eta_p^2 = .04$; Response efficiency: $F(1,28) = 0.95$, $p = 0.34$, $\eta_p^2 = .03$).

We also performed paired-samples *t*-tests (Bonferroni-corrected for four comparisons) comparing colour-change detection for each experimental condition with the letter string baseline (RT: Mean ± *SD* = 0.53 ± 0.17 s; Accuracy: Mean ± *SD* = 74.7% ± 24.5%; Response efficiency: Mean ± *SD* = 0.85 ± 0.57). Responses to baseline trials were significantly faster than for reading or recall trials ($t(28) > 2.84$, $p < 0.009$). For both accuracy and response efficiency, there were no significant differences between the experimental tasks and the letter string baseline data ($t(28) < 1$).

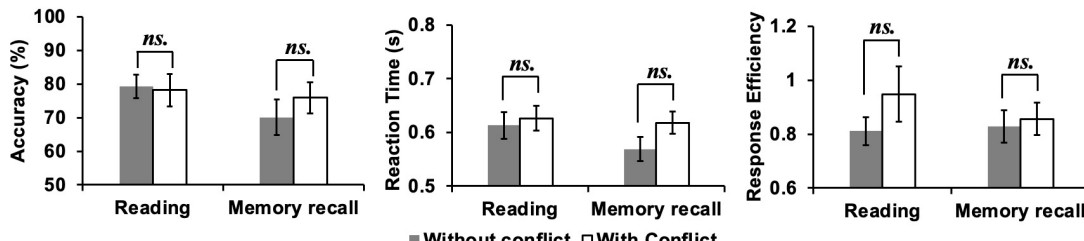

**Appendix 1—figure 1.** Behavioural results of catch trials. Accuracy (percentage correct, left panel) and reaction time (in seconds, middle panel), as well as response efficiency (right panel) for the catch trials in each experimental condition (*Pure Reading*, *Conflict Reading*, *Pure Recall*, and *Conflict Recall*). Error bars represent the standard error. *ns.* indicates *not significant*.

## Effects of each experimental condition over letter string baseline

For reading, the bilateral temporal regions (i.e., temporal poles, superior/middle/inferior temporal gyrus), precentral gyrus, middle/inferior frontal gyrus, temporal fusiform cortex, supplementary motor cortex, and visual cortex showed greater activation relative to the letter string baseline (see *Appendix 1—figure 2A–B* of *Pure reading > Letter baseline* and *Conflict reading > Letter baseline* respectively). For autobiographical memory, middle temporal gyrus, temporal pole, middle/inferior frontal gyrus, insular cortex, supplementary motor cortex, and visual cortex showed greater activation compared to the letter string baseline (see *Appendix 1—figure 2C–D* of the *Pure AM retrieval > Letter baseline* and *Conflict AM retrieval > Letter baseline*).

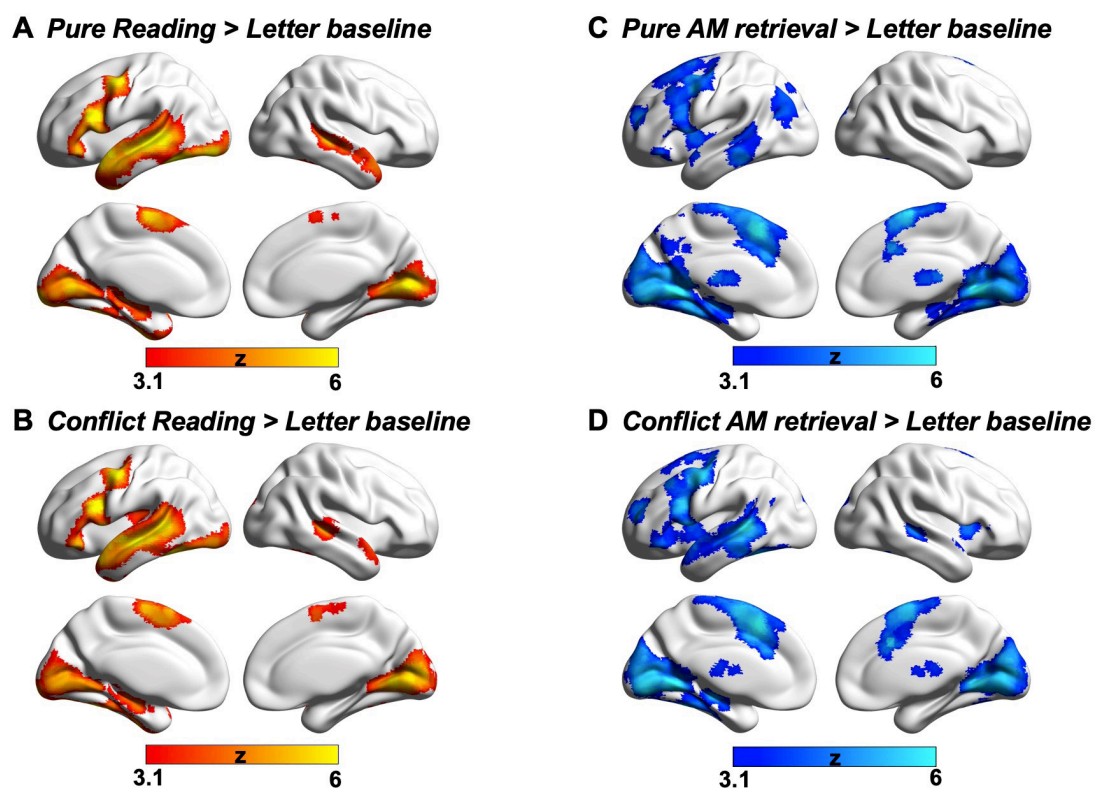

**Appendix 1—figure 2.** Comparisons between each experimental condition and meaningless letter string stimuli processing. All maps were cluster-corrected with a voxel inclusion threshold of *z* > 3.1 and family-wise error rate using random field theory set at *p* < 0.05.

## Activation and deactivation in reading and autobiographical memory retrieval

To identify activation and deactivation elicited by each task, we performed a formal conjunction analysis on the contrast maps of conflict and no-conflict for each experimental condition over the letter string baseline (providing a basic level of control for visual input and button presses). For reading, the bilateral temporal regions (i.e., temporal poles, superior/middle/inferior temporal gyrus), precentral gyrus, middle/inferior frontal gyrus, temporal fusiform cortex, supplementary motor cortex, and visual cortex showed activation relative to the letter string baseline (see *Appendix 1—figure 3A*; the conjunction of *Pure reading > Baseline* and *Conflict reading > Baseline*), while bilateral middle frontal gyrus, supramarginal gyrus, medial prefrontal gyrus, anterior/posterior cingulate gyrus, and precuneus showed deactivation (see *Appendix 1—figure 3B*; the conjunction of *Baseline > Pure reading* and *Baseline > Conflict reading*). For autobiographical memory, middle temporal gyrus, temporal pole, middle/inferior frontal gyrus, insular cortex, supplementary motor cortex, and visual cortex showed activation compared to the letter string baseline (see *Appendix 1—figure 3C*; the conjunction of *Pure AM retrieval > Baseline* and *Conflict AM retrieval > Baseline*), while supramarginal gyrus showed deactivation (see *Appendix 1—figure 3D*; the conjunction of *Baseline > Pure AM retrieval* and *Baseline > Conflict AM retrieval*).

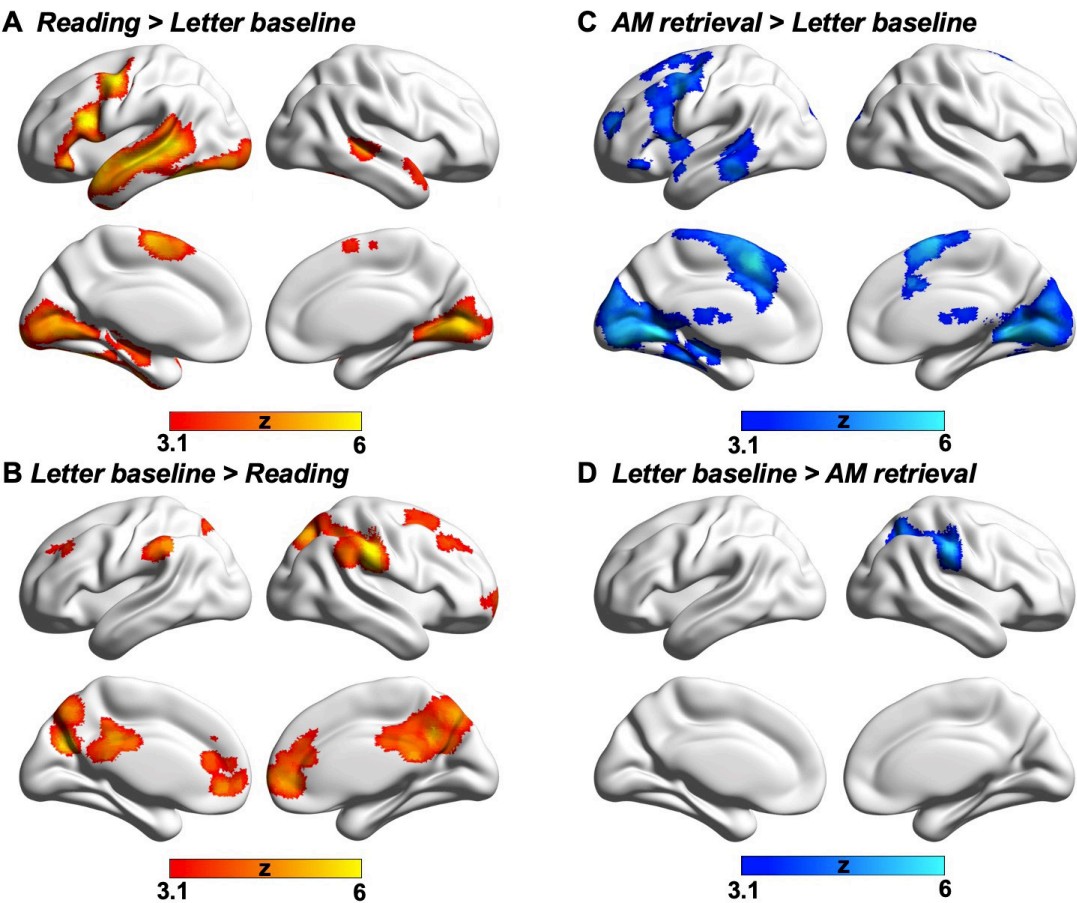

**Appendix 1—figure 3.** Task activation and deactivation. (**A**) Reading > Letter baseline and (**B**) Letter baseline > Reading show the brain activation and deactivation during reading task relative to the letter string baseline. (**C**) AM retrieval > Letter baseline and (**D**) Letter baseline > AM retrieval show the brain activation and deactivation during autobiographical memory recall relative to the letter string baseline. These conjunctions were identified using FSL's 'easythresh_conj' tool. All maps were thresholded at $z > 3.1$ (cluster-size $p$-FWE <0.05).

## Common activation in both reading and autobiographical memory recall tasks

We examined the brain regions activated in both reading comprehension and autobiographical memory recall tasks in a whole-brain analysis. We computed a formal conjunction across the contrast maps of each task over the letter string baseline and established that they overlapped in precentral gyrus, middle/inferior frontal gyrus, temporal pole, middle temporal gyrus, temporal fusiform cortex, supplementary motor cortex, parahippocampus and visual cortex (intracalcarine cortex and lingual gyrus; see *Appendix 1—figure 4A*). The mean percentage signal change of each experimental condition over letter baseline in each cluster is presented in *Appendix 1—figure 4A*. We also compared these regions of overlap with three DMN subsystems – core DMN, dorsal medial subsystem and medial temporal subsystem – defined by *Yeo et al., 2011* in their 17-network parcellation of intrinsic connectivity patterns. Of those voxels which fell within DMN, 95% were within the dorsomedial DMN subnetwork (see *Appendix 1—figure 4B* with pie chart showing the percentage of overlap with each subsystem). In summary, this analysis shows that lateral temporal DMN regions are implicated in both reading comprehension and autobiographical memory retrieval.

**A** Conjunction of Reading and Autobiographical Memory Recall

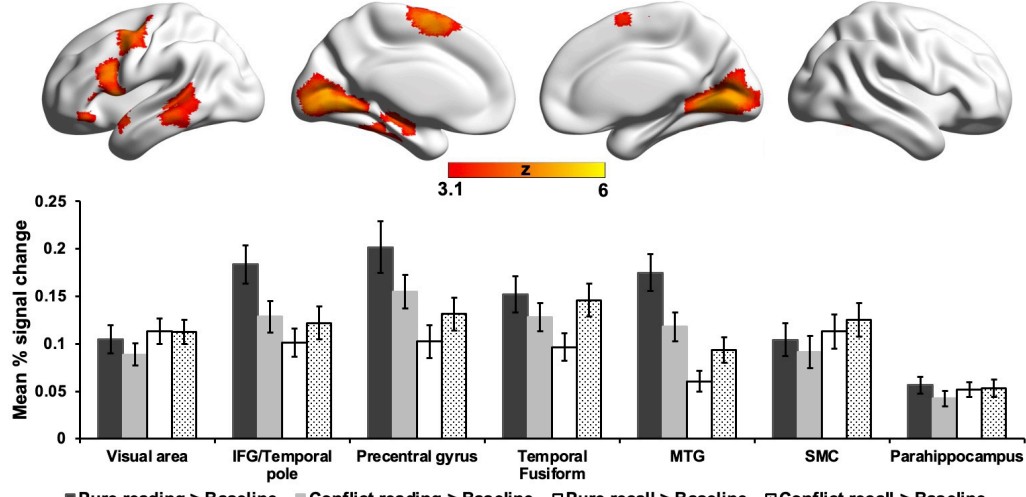

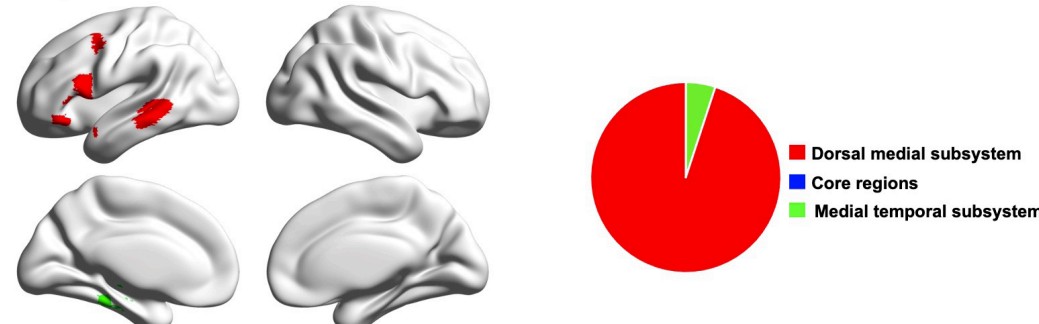

**B** Overlap with default mode network

**Appendix 1—figure 4.** Conjunction analysis. (**A**) Conjunction of brain activation during reading comprehension and autobiographical memory recall, with this conjunction identified using FSL's 'easythresh_conj' tool. The bar chart shows the mean % signal change of each experimental condition over letter baseline in each identified cluster. Error bars represent the standard error. (**B**) The pie chart illustrates the percentages of voxels that were within the DMN in the task conjunction map that overlap with each DMN subsystem: the medial temporal subsystem is shown in green, and the dorsomedial subsystem is shown in red. The DMN conjunction largely fell within the dorsomedial subsystem of DMN. All maps were thresholded at $z > 3.1$ (cluster-size $p$-FWE <0.05). IFG = inferior frontal gyrus; MTG = middle temporal gyrus; SMC = supplementary motor cortex.

## Effects of task conflict

We did not find any conflict effects for reading, over and above the task focus effects we present in the main manuscript. For autobiographical memory, parahippocampal gyrus, temporal occipital fusiform, lateral occipital cortex, precuneus and anterior medial prefrontal cortex – identified as important areas for autobiographical memory retrieval by the *Autobiographical memory > Reading* contrast – showed greater activation when there was *no* distracting sentence input (see ***Appendix 1—figure 5A***). When there was conflict from irrelevant sentences during autobiographical memory recall, there was greater activation in some regions that were identified in the *Autobiographical memory > Reading* contrast, including precentral gyrus, frontal pole, frontal orbital cortex, and angular gyrus, plus greater activation in middle temporal gyrus and ventral visual cortex corresponding to the presentation of more complex visual input (see ***Appendix 1—figure 5B***).

### A  *Pure AM retrieval > Conflict AM retrieval*

### B  *Conflict AM retrieval > Pure AM retrieval*

**Appendix 1—figure 5.** Effects of Task conflict. (**A**) Significant activation when there was no conflict from semantic input defined using the contrast of *Pure AM retrieval > Conflict AM retrieval*. (**B**) Significant activation when there was conflict from semantic input defined using the contrast of *Conflict AM retrieval > Pure AM retrieval*. All maps were thresholded at *z* > 3.1 (cluster-size *p*-FWE <0.05).

## Supporting materials for Experiment 2

### Contents of naturally-occurring mind-wandering during reading

The assessment of naturally-occurring mind-wandering behaviour contained 22 questions about the content of off-task thoughts, rated on a scale of 1 (Completely did not describe my thoughts) to 9 (Completely did describe my thoughts). These questions refer to four types of mind-wandering contents, namely Past thoughts (e.g., *I thought about an event that took place earlier today*), Future thoughts (e.g., *I thought about an interaction I may possibly have in the future*), Social thoughts (e.g., *I thought about people I have just recently met*), and Emotional thoughts (e.g., *I thought about things I am currently worried about*). The descriptive rating for each type of mind-wandering content is shown in *Appendix 1—figure 6*.

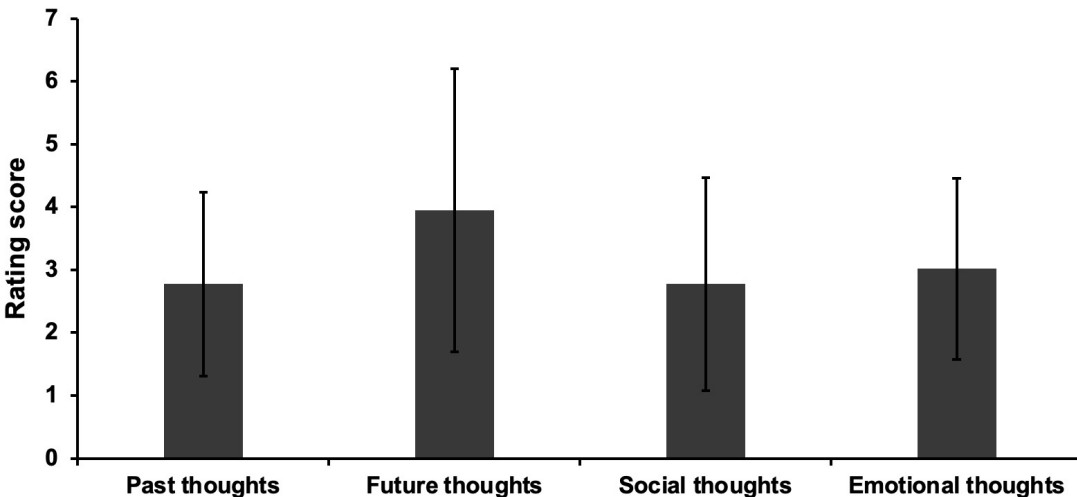

**Appendix 1—figure 6.** Contents of naturally-occurring mind-wandering during reading. Different types of internal thoughts that people reported that they were thinking about, when they mind-wandered during reading in Experiment 2.

### Results of resting-state connectivity analysis using a stringent motion criterion

To examine the stability of the intrinsic connectivity results presented in our main text, we re-ran all the resting-state functional connectivity analyses in Experiment 2 with a more stringent head

motion criterion (i.e., mean framewise displacement > 0.2 mm), which led to the exclusion of 5 participants in Dataset 1 (i.e., N = 239 after removal) and two participants in Dataset 2 (i.e., N = 67 after removal). We found that the resulting patterns of these two datasets remained the same as those found in the main text: in Dataset 1, the DMN seed linked to reading exhibited stronger functional coupling with ventral visual cortex relative to the DMN seed linked to autobiographical memory retrieval (see *Appendix 1—figure 7A*), the 'task-rest' conjunction also establishes that the functional connectivity of the seed linked to reading was mirrored by joint activation of these regions during reading (see *Appendix 1—figure 7B*); For the mind-wandering reading individual difference analysis (i.e., Dataset 2), for individuals with better reading comprehension and less generated mind-wandering contents, there was also greater functional coupling of the DMN seed linked to reading to the regions of left (corrected cluster-size *p*-FWE <0.001) and right primary visual cortex (corrected cluster-size *p*-FWE = 0.002), as well as postcentral gyrus (corrected cluster-size *p*-FWE = 0.005; see *Appendix 1—figure 7C*).

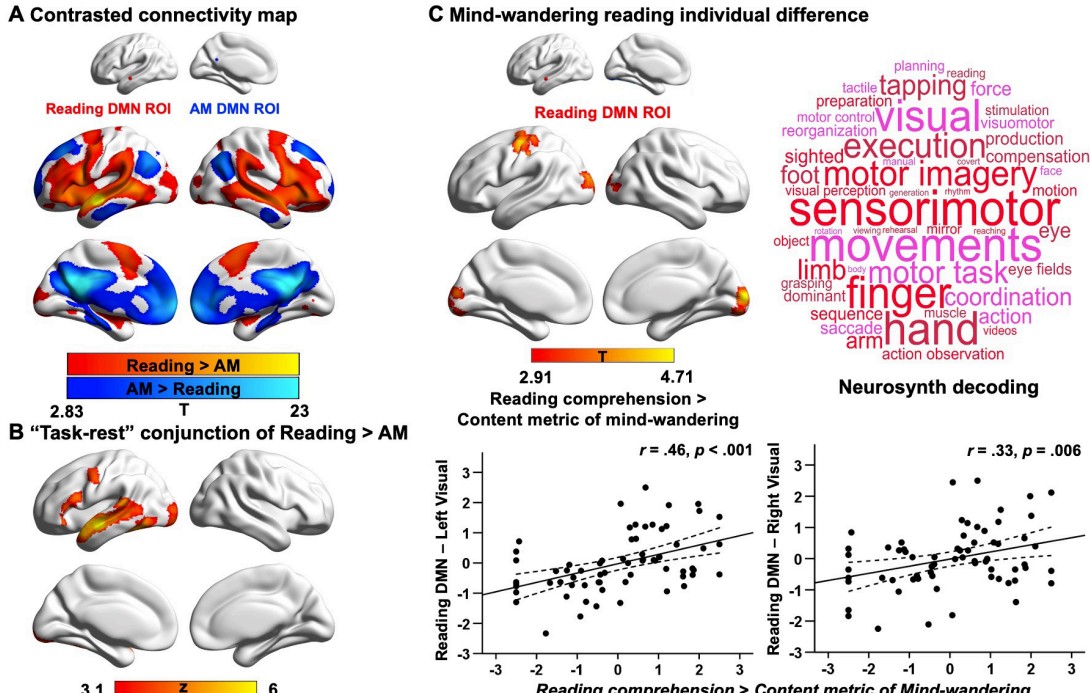

**Appendix 1—figure 7.** Results of resting-state intrinsic connectivity using a stringent motion criterion. Panel (**A**) shows the results of a functional connectivity analysis examining differences between the DMN seeds (reading DMN MNI coordinates: − 56, -10, -12; autobiographical memory DMN MNI coordinates: − 6, -50, 22). Regions showing stronger intrinsic connectivity to the reading DMN seed are shown in warmer colours, while regions showing stronger intrinsic connectivity to the autobiographical memory DMN seed are shown in cooler colours. The lower panel (**B**) shows results of a formal conjunction between regions associated with greater activity during reading versus autobiographical memory retrieval, and regions showing stronger correlation at rest with the DMN seed most activated by reading. Panel (**C**) shows the relationship of these seeds' functional architecture and self-reports of mind-wandering during reading. Group-level regression, using the DMN seed showing peak activation during reading, demonstrated stronger connectivity with the regions of primary visual areas and postcentral gyrus in individuals with good comprehension and less internally-generated mind-wandering content. The scatterplots present the correlation between behaviour and the average correlation with the reading-relevant DMN seed and the identified visual clusters (beta values). The error lines on the scatterplot indicate the 95% confidence estimates of the mean. Each point describes an individual participant. The word cloud shows the functional associations with this connectivity map using Neurosynth.

## Structural connectivity analysis

Our functional connectivity analysis revealed that reading-relevant DMN regions are more functionally connected to ventral visual cortex, compared to regions of DMN relevant to autobiographical memory retrieval. This effect might be due to the differences in the structural connectivity between visual cortex and these DMN subsystems. To examine this possibility, we performed white matter

connectivity analysis using the visual network from *Yeo et al., 2011* as a seed, and reading- and AM-related DMN regions as masks, which allowed us to delineate patterns of structural connectivity from visual cortex to core and dorsomedial DMN subsystems.

To estimate structural connectivity between the visual network and reading- and AM-related DMN regions, we analysed a pre-processed dataset of unrelated healthy young adults (N = 70, mean age = 31.97 ± 8.82 years old, 31 females). All processing was performed via micapipe, an openly accessible processing pipeline for multimodal MRI data (https://micapipe.readthedocs.io/). In brief, micapipe combines procedures from several software packages, including tools from AFNI, FSL, FreeSurfer, mrtrix, and ANTs. Cortical surfaces were generated by applying FreeSurfer to T1w scans (*Dale et al., 1999*), images were non-linearly aligned to MNI152 space using ANTs, and tissue types segmented using FSL FAST. The diffusion weighted imaging (DWI) data were pre-processed using MRtrix (*Tournier et al., 2019*; *Tournier et al., 2012*). DWI data was denoised, underwent b0 intensity normalisation, and were corrected for susceptibility distortion, head motion, and eddy currents using a reverse phase encoding from two b = 0 s/mm2 volumes. Required anatomical features for tractography processing (i.e., FSL-based tissue type segmentations) were co-registered to native DWI space using an affine transformation implemented in ANTs. Diffusion processing and tractography were performed in native DWI space. We performed anatomically-constrained tractography using tissue types (cortical and subcortical grey matter, white matter, cerebrospinal fluid) segmented from each participant's pre-processed T1w images registered to native DWI space. We estimated multi-shell and multi-tissue response functions and performed constrained spherical-deconvolution and intensity normalisation. To approximate ROI-to-ROI structural connectivity, we applied tckgen with a setting of 100,000 streamlines between the visual network and reading/AM components of the DMN in diffusion space, followed by generation of a tract-weighted image. Tract-weighted images were then mapped to MNI152 space using the existing co-registration/registration functions, and averaged across subjects to approximate consistent tracts across individuals. This analysis revealed that there are structural connectivity pathways from visual areas to DMN regions supporting reading and personal memory retrieval. The results are shown in *Appendix 1—figure 8*.

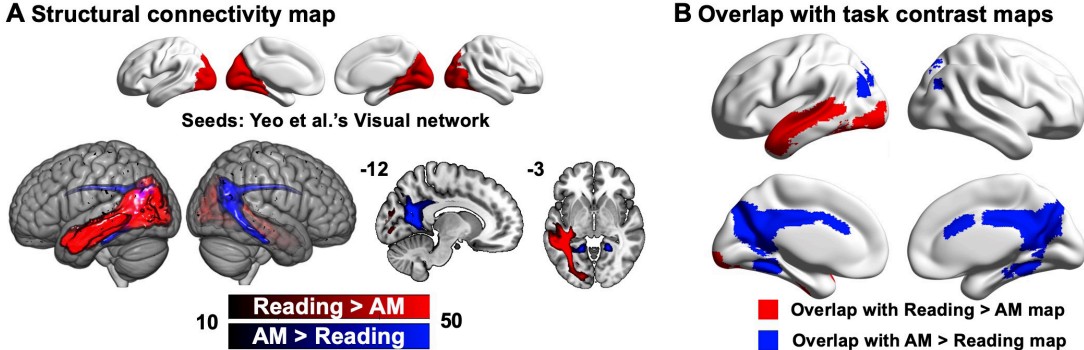

**A Structural connectivity map**

Seeds: Yeo et al.'s Visual network

Reading > AM
AM > Reading

**B Overlap with task contrast maps**

Overlap with Reading > AM map
Overlap with AM > Reading map

**Appendix 1—figure 8.** Structural connectivity. (**A**) Structural connectivity seeding from visual network defined by *Yeo et al., 2011* to DMN regions linked to reading comprehension (in red) and autobiographical memory retrieval (in blue). (**B**) Overlap of structural connectivity map with *Reading > AM* and *AM > Reading* task contrast maps.

