## [Editor Report]

This manuscript is of broad interest to those interested in the relationship between mind-wandering and reading, at the behavioural and neural levels, including when both processes occur at the same time. As such, this manuscript has important implications for clarifying how the experience of mind-wandering while reading may occur.

---

## [Decision Letter]

**Decision letter after peer review:**

Thank you for submitting your article "Perceptual coupling and decoupling of the default mode network during mind-wandering and reading" for consideration by *eLife*. Your article has been reviewed by 2 peer reviewers, and the evaluation has been overseen by a Reviewing Editor and Chris Baker as the Senior Editor. The following individual involved in review of your submission has agreed to reveal their identity: Jessica Andrews-Hanna (Reviewer #1).

Essential revisions:

1. The details of the Experiment 1 design are not fully clear when the Experiment 1 results are presented and Figure 1 is introduced. It could increase readability to provide an overview of the task before describing the results. It would also be helpful to include some details of the analysis in the main text, such as whether trials with color changes were included in analyses.

2. The motion exclusion criteria for Experiment 2 datasets 1 and 2 was >.4mm mean frame-to-frame displacement (I believe after censoring). This is high, as >.2 mm or >.15 mm is more typical. Are results consistent when more stringent motion criteria are applied? Related to this, because motion can be a significant confound in functional connectivity studies, did the individual differences analysis account for differences in head motion between individuals?

3. Although the whole-brain, data-driven analysis approaches used in both experiments have benefits, they make connections between the studies somewhat challenging to follow. For example, it is not readily apparent how precisely the findings agree across datasets because the regions described in each dataset partially but do not fully overlap, and it's not clear what degree of overlap should be considered a replication or what the likelihood of seeing that degree of overlap by chance would be. It could be a stronger, clearer test of the hypotheses proposed to use the same regions of interest across studies. For example, do the same regions that show greater activity during reading than memory show strong connectivity in Experiment 2 dataset 1 and do those same connections scale with propensity to mind wandering and reading comprehension in Experiment 2 dataset 2?

4. Related to the point above, Experiment 2 data show strong resting-state functional connectivity between DMN regions linked to reading and visual ROIs, and weaker functional connectivity between DMN regions linked to memory and these visual ROIs. Are these same functional connections modulated by task condition in Experiment 1? It would strengthen the argument that DMN-visual cortex coupling is disrupted by mind wandering during reading to show that the exact same ROIs are more strongly functionally connected during reading than memory retrieval.

5. The individual differences analysis in Experiment 2 dataset 2 is very interesting but doesn't provide conclusive support for the hypothesis that decoupling between the DMN and visual regions is causally related to mind wandering. Rather, the result suggests that the strength of a functional connection within the DMN scales with self-reported mind wandering. A more direct test of the hypothesis proposed in the introduction would be relating Experiment 2 dataset 2 participants' functional connectivity between DMN and visual ROIs (the same ROIs defined in Experiment 1) to their self-reported mind wandering. The proposed theory should predict a relationship between greater DMN-visual ROI connectivity, less propensity to mind wander, and better reading comprehension. Even in the absence of a more direct test of the hypothesis, it would strengthen the observed result if it were replicated in subsets of the data or a new dataset given the exploratory nature of the analysis.

6. The results from Experiment 2's analysis of dataset 2 overlap with findings from Zhang et al., (2019, Scientific Reports) using the same dataset. Although the fact that these data were previously analyzed is helpfully noted in the text, it should also be clarified that paper related individual differences in DMN-visual cortex functional connectivity to individual differences in mind wandering during reading. Thus the analysis of this dataset in the current manuscript is very similar to this previously published analysis. The distinction between these analyses should be clarified in the text.

7. The manuscript would benefit from more discussion of the many discrepancies between mind wandering during naturalistic reading and the authors' current design. Also, in light of these discrepancies, the conclusions (while possibly accurate) do not seem to be directly supported by the data and should be discussed in a more tentative way.

8. The authors should discuss the null "conflict" findings in the discussion, and also speculate as to why they did not observe positive relationships with task focus in regions related to reading. Additionally, it would be helpful if the authors were to specify which task contrasts were used to create Figure 3C.

9. The manuscript often describes separate DMN subdivisions that support reading versus autobiographical memory, but glosses over possible similarities between the two tasks. I would like to see the authors raise the issue of possible neural similarities between the two tasks in the discussion, and clarify that when discussing neural differences, such differences emerged from direct contrasts between the two tasks.

10. I would like to see the authors cite earlier meta-analyses, such as those by Binder et al., 2009 and Yang et al., 2019. In addition, other published work linking reading to the DMN and the dorsal medial subsystem seems important to cite as well, including Andrews-Hanna, Smallwood and Spreng, 2014 who noted a link to reading in their Neurosynth meta-analysis of default mode subsystems, as well as a more recent task-related meta-analysis quantifying the involvement of the dorsal medial subsystem (Yang et al., 2019).

11. I would like to see more discussion surrounding the resting state individual difference relationships between mind-wandering during reading and connectivity with the dorsal occipital cortex. Since this region was activated in the autobiographical memory task, I suspect the authors hypothesized the opposite relationship as to that which they observed.

12. Finally, I would like to see more detail presented on the nature of the expository text itself. Perhaps the authors could include a few example stimuli in the manuscript? Are the sentences neutral in valence? Is the text interesting to most participants, or was it meant to be boring? Also, I can't seem to find how long the duration of the memory retrieval and reading components of the trials were, on average. Could the authors add this information into the methods?

Relevant References:

Andrews-Hanna, J. R., Smallwood, J., and Spreng, R. N. (2014). The default network and self-generated thought: component processes, dynamic control, and clinical relevance. Annals of the New York Academy of Sciences, 1316, 29-52. https://doi.org/10.1111/nyas.12360

Binder, J. R., Desai, R. H., Graves, W. W., and Conant, L. (2009). Where is the semantic system ? A critical review and meta-analysis of 120 functional neuroimaging studies. Cerebral Cortex, 19(12), 2767-2796. https://doi.org/10.1093/cercor/bhp055

Yang, X. H., Li, H. J., Lin, N., Zhang, X. P., Wang, Y. S., Zhang, Y., Zhang, Q., Zuo, X. N., and Yang, Y. F. (2019). Uncovering cortical activations of discourse comprehension and their overlaps with common large-scale neural networks. NeuroImage, 203(September), 116200. https://doi.org/10.1016/j.neuroimage.2019.116200

---

## [Author Response]

Essential revisions:1. The details of the Experiment 1 design are not fully clear when the Experiment 1 results are presented and Figure 1 is introduced. It could increase readability to provide an overview of the task before describing the results. It would also be helpful to include some details of the analysis in the main text, such as whether trials with color changes were included in analyses.

We have now provided an overview of Experiment 1’s design before the description of the results. For the task-based fMRI data analysis, we included all the reading and recall trials of each experimental condition – i.e., including the colour-change catch trials, since there were no behavioural differences in the detection rates across conditions and the effect of these trials was explicitly accounted for in the general linear model during analysis. We have included more details about the task-based fMRI analyses in this revision. Please see below for further details.

Overview of task design of Experiment 1 on Page 5-6:

“Design

The experiment took place over two days. On Day 1, participants were asked to identify specific personal events linked to each autobiographical memory cue (words like PARTY). […] After each reading or memory recall trial, task focus ratings on a scale of 1 (i.e., not at all) to 7 (i.e., very much) were collected to index the extent to which participants were able to focus on the primary task.”

Overview of task-based fMRI data analysis of Experiment 1 on Page 8:

“Neuroimaging results: Having established the expected pattern of competition between autobiographical memory retrieval and reading, we next considered the neural correlates that distinguish these states. […] All the reading and memory recall trials were included in the analysis – including the rare catch trials with colour changes, since there were no behavioural differences in colour-change detection rates across these conditions (see Appendix 1-Figure 1) and these catch trials were explicitly modelled in our analysis.**”**

2. The motion exclusion criteria for Experiment 2 datasets 1 and 2 was >.4mm mean frame-to-frame displacement (I believe after censoring). This is high, as >.2 mm or >.15 mm is more typical. Are results consistent when more stringent motion criteria are applied? Related to this, because motion can be a significant confound in functional connectivity studies, did the individual differences analysis account for differences in head motion between individuals?

In Experiment 2, CONN was used in our resting-state functional connectivity analysis, which is a standard and widely-used toolbox for functional connectivity analysis. To control the impact of head motion, factors that are identified as potential confounding effects to the estimated BOLD signal are estimated and removed separately for each voxel and for each subject. This means that, except for the removal of participants with excessive head motion, another two confounding effects related to head motion were also included to account for its negative influence: (i) one was estimated subject-motion parameters (Friston et al., 1995) in order to minimize motion related BOLD variability, (ii) another one was identified outlier scans or scrubbing (Power et al., 2014) to remove any influence of these outlier scans on the BOLD signal. These potential confounding effects were included as nuisance parameters into the model in the denoising step. CONN uses the standard CompCor approach (Behzadi, Restom, Liau, and Liu, 2007), removing all of these effects to obtain a clean signal for the following analyses. Global signal regression was not performed because CompCor can effectively account for subject movement effects and other sources of noise in the BOLD signal (Behzadi et al., 2007; Muschelli et al., 2014). In this way, our group-level analysis which was performed based on the pre-processed and denoised signal has accounted for both motion and other sources of noise. We have further clarified these aspects in the Methods.

In addition, we have re-run all the resting-state functional connectivity analyses in Experiment 2 using a more stringent head motion criterion as you suggested (i.e., mean framewise displacement >.2 mm), leading to the exclusion of 5 participants in Experiment 2 Dataset 1 (N = 239 after removal) and 2 participants in Dataset 2 (N = 67 after removal). The resulting patterns of these two datasets remained the same as when more lenient criteria for head motion were used: In Experiment 2 Dataset 1, the DMN seed linked to reading exhibited stronger functional coupling with ventral visual cortex relative to the DMN seed linked to autobiographical memory retrieval (see Appendix 1-Figure 7 A) and the “task-rest” conjunction also established that the functional connectivity of the seed linked to reading is mirrored by their joint activation during reading (see Appendix 1-Figure 7 B). For the mind-wandering during reading individual difference analysis, for individuals with better reading comprehension and less generated mind-wandering contents, there was also greater functional coupling of the DMN seed linked to reading to the regions of left (corrected cluster-size p-FWE <.001) and right primary visual cortex (corrected cluster-size p-FWE = .002), as well as postcentral gyrus (corrected cluster-size p-FWE = .005; see Appendix 1-Figure 7 C). We have presented the results using these stricter exclusion criteria in the Appendix 1.

Clarification of the control of head motion in the Methods section on Page 27:

“Pre-processing steps automatically created three first-level covariates: a realignment covariate containing the six rigid-body parameters characterising the estimated subject motion for each participant, a scrubbing covariate containing the potential outliers scans for each participant (i.e., identified through the artefact detection algorithm included in CONN, with intermediate settings: scans for each participant were flagged as outliers based on scan-by-scan change in global signal above z = 5, subject motion threshold above 0.9 mm, differential motion and composite motion exceeding 97% percentile in the normative sample), and a covariate containing quality assurance (QA) parameters (e.g., the global signal change from one scan to another and the framewise displacement) for each participant. […] Global signal regression was not performed because CompCor can account for subject movement effects and other sources of noise in the BOLD signal (Behzadi et al., 2007; Muschelli et al., 2014).”

Note of the presentation of the results of connectivity analysis using stricter exclusion criteria in the Appendix 1 on Page 14:

**“**We re-ran all of these analyses using a more stringent motion criterion, excluding individuals with mean head motion that exceeded 0.2 mm. The same pattern of results was obtained. These results are presented in the Appendix 1.**”**

3. Although the whole-brain, data-driven analysis approaches used in both experiments have benefits, they make connections between the studies somewhat challenging to follow. For example, it is not readily apparent how precisely the findings agree across datasets because the regions described in each dataset partially but do not fully overlap, and it's not clear what degree of overlap should be considered a replication or what the likelihood of seeing that degree of overlap by chance would be. It could be a stronger, clearer test of the hypotheses proposed to use the same regions of interest across studies. For example, do the same regions that show greater activity during reading than memory show strong connectivity in Experiment 2 dataset 1 and do those same connections scale with propensity to mind wandering and reading comprehension in Experiment 2 dataset 2?

Thank you for your helpful suggestion. We have now reported functional connectivity analyses using the same regions of interest across studies to show the extent to which patterns are replicated across datasets. In Experiment 2 Dataset 1, the reading and autobiographical memory DMN peak activation sites from Experiment 1 were used as seeds in an analysis of intrinsic connectivity, since these locations represent the maximal differential activation relating to each specific task state. Using these seeds, we performed a whole-brain seed-to-voxel connectivity analyses plus an ROI-based analysis to examine whether there was difference in the connectivity of these DMN nodes to the visual cluster that was more engaged in the reading task in Experiment 1. In this way, we can maintain the benefits of data-driven analyses while making it easier to relate the findings across datasets. In Experiment 2 Dataset 2, we examined the relationship between the intrinsic connectivity of the same DMN seeds (derived from Experiment 1) with individual variance in naturalistic mind-wandering during reading. This also recovered a cluster in visual cortex. Following your other suggestion below, we then examined functional connectivity in a task context (using data from Experiment 1) to establish whether coupling between the DMN seeds and the mind-wandering visual cluster from Experiment 2 was reduced when participants focussed on autobiographical memory. The results of these new analyses are summarised below.

For Experiment 2 Dataset 1, we found, as expected, that the DMN peak seed linked to reading showed greater intrinsic connectivity to ventral visual cortex than the DMN seed associated with autobiographical memory retrieval. A follow-up ROI analysis revealed that there was stronger connectivity at rest between Reading DMN-to-Reading visual areas than Autobiographical memory DMN-to-Reading visual areas. In this analysis, we used a Reading visual ROI (i.e., a sphere based on the peak visual activation in the task contrast of Reading > Autobiographical memory retrieval, as shown in Figure 2A; MNI coordinates: -32, -92, -10) and extracted functional connectivity for each participant, for each DMN seed. Paired samples *t*-tests revealed stronger intrinsic connectivity between the Reading DMN and Reading visual cortex than the Autobiographical memory DMN to Reading visual cortex (Reading DMN-to-Reading visual: Mean ± SD = .02 ±.14, Autobiographical memory DMN-to-Reading visual: Mean ± SD = -.01 ±.14; *t*(242) = 2.78, *p* = .006), providing further evidence that the same regions showing stronger activation during reading comprehension than memory retrieval also exhibit stronger intrinsic connectivity at rest.

For the results of Experiment 2 Dataset 2, we found that individuals with more off-task thought contents and poorer comprehension had weaker connectivity between the DMN seed linked to reading and primary visual cortex. This pattern is informative since it suggests that perceptual decoupling is related to individual variation in mind-wandering during reading; importantly, it generalises our experimental paradigm to an ecologically valid situation. To better understand whether this identified “mind-wandering-related” visual cluster behaves in a similar way during the experimentally-mimicked experiences of mind-wandering reading, we then performed psychophysiological interaction (PPI) analysis (i.e., task-based functional connectivity) using the Reading and Autobiographical memory DMN seeds. For each participant, the functional connectivity of each seed with this “mind-wandering” visual ROI (created based on peak visual activation in Experiment 2 Dataset 2; see Figure 4C) was extracted for each experimental condition, as well as the Task Focus effect of each experimental condition (i.e., the parametric regressors of Task Focus). This analysis revealed that the functional decoupling of Reading DMN to this “mind-wandering-related” visual site was greater (i) during the retrieval of personally-relevant memories in the absence of sentences, and (ii) when the task focus was higher on memory retrieval but when there was conflict from irrelevant semantic inputs. The similar perceptual decoupling pattern in both resting-state and task-state functional connectivity highlights the importance of perceptual decoupling when internal memory retrieval competes with reading comprehension.

Taken together, this converging lines of evidence from Experiment 2 suggest that (i) the generation of internal thoughts relies on DMN regions that functionally decouple from visual areas important for reading, and (ii) this consequently creates a perceptually-decoupled mental state that disrupts ongoing reading comprehension. We have updated all of these results in this revision. Please see below for details.

The results of these new analyses in Experiment 2 on Page 11-17:

“Experiment 2

Experiment 1 established that our paradigm captured the expected mutual inhibition between reading and autobiographical memory retrieval (Figure 1) and found that both states depend on activity within distinct regions within the DMN (Figure 2 and 3). […] Importantly, this analysis establishes a similar perceptual decoupling pattern for this visual site in both naturally-occurring and experimentally-mimicked experiences of mind-wandering during reading, highlighting the importance of perceptual decoupling when memory retrieval competes with reading.”

Revised Discussion on Page 19:

**“**Moreover, individual differences in intrinsic connectivity associated with the state of mind-wandering during reading are consistent with this view. […] These converging lines of evidence demonstrate the importance of perceptual decoupling for mind-wandering during reading, which consequently disrupts the pattern of perceptual coupling that supports reading comprehension (Smallwood, 2011).**”**

4. Related to the point above, Experiment 2 data show strong resting-state functional connectivity between DMN regions linked to reading and visual ROIs, and weaker functional connectivity between DMN regions linked to memory and these visual ROIs. Are these same functional connections modulated by task condition in Experiment 1? It would strengthen the argument that DMN-visual cortex coupling is disrupted by mind wandering during reading to show that the exact same ROIs are more strongly functionally connected during reading than memory retrieval.

Thank you for your valuable suggestion. As mentioned above, we have performed PPI analysis to examine whether DMN-to-Visual decoupling is modulated by our task manipulations using the Reading/Autobiographical memory DMN peak seeds. To extract DMN-to-Visual connectivity values, we created the visual ROI based on the peak visual activation identified in Experiment 2 Dataset 2 (as Figure 4C). This visual site is associated with individual differences in naturally-occurring mind-wandering during reading. The exploration of the decoupling from this “mind-wandering” relevant visual site would allow us to examine to what extent this pattern behaves in a similar way in the experiment simulating mind-wandering during reading. The results suggested that there is greater Reading DMN-to-Visual decoupling during the retrieval of autobiographical memories in the absence of semantic inputs and when task focus is greater for internally-generated thoughts instead of irrelevant semantic inputs. These findings support our argument that the functional coupling of reading-related DMN and visual cortex is disrupted by mind-wandering since the retrieval of autobiographical mental content creates a perceptually decoupled state that disrupts ongoing reading comprehension.

In this analysis, two PPI models were established using the Reading and Autobiographical memory DMN seeds to estimate the task-based functional connectivity of these regions in DMN with visual cortex in each experimental condition. An ROI-based approach was applied to extract the functional connectivity of Reading/Autobiographical memory DMN seeds to the “mind-wandering” visual ROI for each experimental condition (i.e., main effects of task), as well as for the parametric effect of Task Focus in each condition, for each participant. For each seed model, a 2 (Task: Reading vs. Autobiographical memory) by 2 (Conflict: No conflict vs. Conflict) repeated-measures Analysis of Variance (ANOVA) was performed to examine the differences in functional connectivity across conditions.

For the Reading DMN seed, and examining main effects of condition, the effect of Task approached significance, *F*(1,27) = 3.51, *p* = .072, *η*_*p*_^*2*^ = .12. There was also a significant interaction effect between Task and Conflict, *F*(1,27) = 5.42, *p* = .028, *η*_*p*_^*2*^ = .17. Simple effects tests revealed a significant main effect of Task in the absence of conflict, suggesting stronger functional coupling of Reading DMN to visual areas during reading comprehension. For the parametric effects of Task Focus, there were no significant main effects of either Task, *F*(1,27) = .61, *p* = .44, *η*_*p*_^*2*^ = .02, or Conflict, *F*(1,27) = .19, *p* = .67, *η*_*p*_^*2*^ = .007. There was a significant interaction effect between these two factors, *F*(1,27) = 6.54, *p* = .016, *η*_*p*_^*2*^ = .20. Simple effects tests revealed a significant main effect of Task in the conflict condition, suggesting greater reading DMN-to-Visual coupling when participants were more focussed on reading comprehension despite distracting autobiographical memory cues. No effects were found for the Autobiographical memory DMN seed model. All of these additional PPI analyses are described in the Results section of Experiment 2, for the details of updated PPI results please see responses to point 3. We have also added a section to explain how these PPI models were set up in the Methods section.

The explanation of the setup of PPI models in the Methods section on Page 29:

“Psychophysiological interaction (PPI) analysis in Experiment 2: In order to understand whether the perceptual decoupling pattern at rest related to individual differences in mind-wandering identified in Experiment 2 also emerges in experimentally-mimicked mind-wandering during reading, we conducted PPI analysis. […] The four sequential runs were combined using fixed-effects analyses for each participant, which allowed us to extract the connectivity parameters for each experimental condition for each participant in each seed model.**”**

5. The individual differences analysis in Experiment 2 dataset 2 is very interesting but doesn't provide conclusive support for the hypothesis that decoupling between the DMN and visual regions is causally related to mind wandering. Rather, the result suggests that the strength of a functional connection within the DMN scales with self-reported mind wandering. A more direct test of the hypothesis proposed in the introduction would be relating Experiment 2 dataset 2 participants' functional connectivity between DMN and visual ROIs (the same ROIs defined in Experiment 1) to their self-reported mind wandering. The proposed theory should predict a relationship between greater DMN-visual ROI connectivity, less propensity to mind wander, and better reading comprehension. Even in the absence of a more direct test of the hypothesis, it would strengthen the observed result if it were replicated in subsets of the data or a new dataset given the exploratory nature of the analysis.

Thank you very much for your suggestions. As we mentioned in the responses to point 3, we found that the functional decoupling between DMN linked to reading and primary visual area is associated with individual differences in mind-wandering during reading when using the Reading/Autobiographical memory peak seeds. Particularly, there is greater reading DMN-to-visual coupling for individuals with better reading comprehension and less generated contents of off-task thoughts (i.e., mind-wandering content metric, including autobiographical memory). This finding provides more conclusive support for our argument that mind-wandering elicits a perceptual decoupling state that disrupts ongoing reading comprehension. Importantly, this finding establishes a generalisation of our results from experimentally manipulated experiences of mind-wandering in reading: similar effects occur during a more naturalistic experience in the laboratory, showing our experimental manipulation mimics key processes engaged during mind-wandering in reading.

6. The results from Experiment 2's analysis of dataset 2 overlap with findings from Zhang et al., (2019, Scientific Reports) using the same dataset. Although the fact that these data were previously analyzed is helpfully noted in the text, it should also be clarified that paper related individual differences in DMN-visual cortex functional connectivity to individual differences in mind wandering during reading. Thus the analysis of this dataset in the current manuscript is very similar to this previously published analysis. The distinction between these analyses should be clarified in the text.

We have now clarified that we also found a DMN-to-Visual decoupling effect associated with individual differences in mind-wandering experience while reading in our previous study (Zhang et al., 2019). In our current analysis, we performed a similar analysis to this previous study, except that we used the Reading/Autobiographical memory DMN peak sites as seeds to drive these functional connectivity analyses. Though both our current Reading DMN and previous DMN seeds are located in the left lateral temporal DMN regions, they are in different locations. The findings from our current study are also consistent with our previous findings, since we found that there was greater decoupling of reading-related DMN from primary visual cortex for individuals reporting more off-task thought contents and poorer reading comprehension. These converging lines of evidence demonstrate the importance of reductions of reading-related DMN-to-Visual coupling for the occurrence of mind-wandering during reading. We have discussed these points in the Results section in this revised manuscript.

Revised text on Page 13-14:

**“**Our second analysis examined how the functional architecture of these DMN seeds associated with reading and autobiographical memory retrieval was related to individual variation in naturally occurring mind-wandering during reading as measured in our prior study (Zhang et al., 2019). […] When seeding the reading DMN peak, the contrast of reading comprehension over mind-wandering content revealed regions of left primary visual cortex (corrected cluster-size p-FWE = .004) and right primary visual cortex (corrected cluster-size p-FWE = .004), as well as postcentral gyrus (corrected cluster-size p-FWE = .006), which showed stronger connectivity for individuals with better reading comprehension and less mind-wandering content reported in a questionnaire (see Figure 4C), in line with our previous findings (Zhang et al., 2019).**”**

7. The manuscript would benefit from more discussion of the many discrepancies between mind wandering during naturalistic reading and the authors' current design. Also, in light of these discrepancies, the conclusions (while possibly accurate) do not seem to be directly supported by the data and should be discussed in a more tentative way.

In this revision we have more carefully evaluated the similarities and differences between the results found in naturalistic reading and in our experimental study. Although we found similar connectivity patterns in these contexts, some important differences remain. First, the reading task in our current paradigm was presented in a fixed-paced manner, and participants read short factual-based text. Naturalistic reading involves a self-paced process, longer passages and a much more complex pattern of attentional focus on the text, with the pace sometimes slowing down or speeding up as one's attention, interest, or the complexity of the text waxes and wanes. In addition, people also often go back to earlier parts of text during reading. Moreover, mind-wandering occurs spontaneously in naturalistic reading which is difficult to capture, and for this reason we pitted these two mental states in competition, deliberately probing its occurrence during reading comprehension. We have now discussed these points as limitations in the Discussion section, however, it is important to note that a unique feature of our study is that we were able to generalise from constrained tasks in the scanner, to a more naturalistic situation.

Revised Discussion on Page 21-22:

“Second, differing from our current design, one’s naturalistic reading is self-paced and involves longer passages with a complex pattern of attentional focus on the text. People sometimes slow down or speed up their reading, as attention, interest or complexity wax and wane, and often, they re-read earlier parts of text. Though there are important discrepancies between the mind-wandering occurring during naturalistic reading and the design of Experiment 1, our study established important neural similarities between task-induced autobiographical memory (Experiment 1) and naturally occurring mind-wandering (Experiment 2) during reading. These similarities cannot be accounted for by differences between the experimental situations and therefore our study establishes important support for the process-occurrence view of self-generated states (Smallwood, 2013).”

8. The authors should discuss the null "conflict" findings in the discussion, and also speculate as to why they did not observe positive relationships with task focus in regions related to reading. Additionally, it would be helpful if the authors were to specify which task contrasts were used to create Figure 3C.

For the conflict effect, we did not find any conflict effects for reading, also no significant parametric effect of Task Focus in the reading-related DMN regions. The absence of these effects might relate to the specific functional role of these reading-related regions in semantic processing. This is because, for the effect of Conflict in the memory retrieval task (i.e., autobiographical memory retrieval with Conflict > autobiographical memory retrieval with no conflict), we found stronger activation in the regions that were identified as key sites for reading comprehension by the Reading > autobiographical memory retrieval contrast (see Appendix 1-Figure 5). This finding suggests that the lateral temporal DMN, along the ventral visual cortex, are more perceptually coupled, with stronger activation in response to meaningful visual inputs (sentences versus letter strings in autobiographical memory retrieval conditions), regardless of whether the perceptual input is relevant to the ongoing task or not. Consequently, these regions may be able to detect situations in which meaning emerges in the environment, even when the focus of attention is elsewhere. In other words, its responsiveness to external meaningful inputs might be more automatic and effortless, without a demanding control of attentional allocation. Therefore, we did not see any conflict or Task focus effect in these regions. We have further discussed that in this revision.

In addition, the effects of Task Focus identified in Figure 3C were relative to the implicit baseline (i.e., the first fixation interval across trials). We have also clarified that in the Figure caption now.

Further Discussion of null conflict effects on Page 21:

“In addition, we found that the regions important for reading comprehension activate not only when visual input is task-relevant (i.e., during reading for comprehension), but also when this input is irrelevant to the ongoing task (i.e., revealed by autobiographical memory retrieval with Conflict > autobiographical memory retrieval with no conflict; see Appendix 1-Figure 5), and irrespective of task focus. In line with our findings, these reading-relevant regions may be more perceptually-coupled supporting visual to conceptual knowledge mapping. In this way, they might be sensitive to situations in which meaning emerges in the external environment, even when the focus of attention is elsewhere.”

Clarification of the baseline for Figure 3C in figure caption:

“(C) Task focus effects in reading (red; negative correlation with task focus) and autobiographical memory recall task (blue; positive correlation with task focus), relative to the implicit baseline (i.e., the first fixation interval).”

9. The manuscript often describes separate DMN subdivisions that support reading versus autobiographical memory, but glosses over possible similarities between the two tasks. I would like to see the authors raise the issue of possible neural similarities between the two tasks in the discussion, and clarify that when discussing neural differences, such differences emerged from direct contrasts between the two tasks.

We did examine the brain regions activated in both reading comprehension and autobiographical memory retrieval tasks. We computed a formal conjunction across the contrast maps of each task over the letter string baseline and established that reading and autobiographical memory retrieval elicited activation in the dorsal medial DMN subsystem, associated with semantic cognition (Badre et al., 2005; Jefferies 2013; Lambon Ralph et al., 2017), as well as in visual regions. Previous studies have shown the engagement of lateral temporal cortex in both reading and memory recall (Svoboda et al., 2006; Ferstl et al., 2008; Summerfield et al., 2009; Andrews-Hanna, Saxe, et al., 2014; Lambon Ralph et al., 2017), since both of these two tasks involve semantic retrieval (e.g., Dehaene, Cohen, Morais, and Kolinsky, 2015; Graham, Lee, Brett, and Patterson, 2003; Spitsyna, Warren, Scott, Turkheimer, and Wise, 2006; Svoboda et al., 2006). The task overlap in these DMN regions was left-lateralised, in line with expectations for a semantic retrieval network (Noonan et al., 2013; Hurley et al., 2015; Rice et al., 2015; Gonzalez Alam et al., 2019; Jackson 2020). We have discussed the commonalities in brain activation between these two tasks in the Discussion section now, and have also clarified that the neural differences emerging from direct contrasts of these tasks. In addition, we have also presented the results of our conjunction analysis of Reading and Autobiographical memory in the Appendix 1.

Further Discussion of commonalities in brain activation between these two tasks on Page 20-21:

“Although different aspects of the DMN play distinctive roles in reading comprehension and autobiographical memory retrieval, there are still some neural similarities between these two mental states; since the lateral temporal and frontal regions within dorsomedial DMN activate in response to both tasks (i.e., identified by a formal conjunction across the contrast maps of each task over the letter string baseline; see Appendix 1-Figure 4). Previous studies have shown the engagement of lateral temporal and inferior frontal cortex in both reading and memory recall (Andrews-Hanna, Saxe, and Yarkoni, 2014; Ferstl, Neumann, Bogler, and Von Cramon, 2008; Lambon Ralph et al., 2017; Summerfield, Hassabis, and Maguire, 2009; Svoboda, McKinnon, and Levine, 2006; Yang et al., 2019). Given that both tasks involve semantic cognition (e.g., Dehaene, Cohen, Morais, and Kolinsky, 2015; Graham, Lee, Brett, and Patterson, 2003; Spitsyna, Warren, Scott, Turkheimer, and Wise, 2006; Svoboda et al., 2006), this finding is consistent with prior work implicating lateral temporal cortex and inferior frontal gyrus in the representation and retrieval of heteromodal conceptual knowledge (Badre, Poldrack, Paré-Blagoev, Insler, and Wagner, 2005; Binder et al., 2009; Jefferies, 2013; Lambon Ralph et al., 2017; Noonan, Jefferies, Visser, and Lambon Ralph, 2013).”

Clarification of the neural differences emerging from direct task contrasts on Page 18:

“Using fMRI to index brain activity, we identified that these two states differentially recruited different aspects of the DMN in direct contrasts of these two tasks, with greater recruitment of the dorsomedial DMN subnetwork when participants were reading for comprehension, and greater activity within the core of the DMN during autobiographical memory retrieval (See also Chiou, Humphreys, and Lambon Ralph, 2020).”

10. I would like to see the authors cite earlier meta-analyses, such as those by Binder et al., 2009 and Yang et al., 2019. In addition, other published work linking reading to the DMN and the dorsal medial subsystem seems important to cite as well, including Andrews-Hanna, Smallwood and Spreng, 2014 who noted a link to reading in their Neurosynth meta-analysis of default mode subsystems, as well as a more recent task-related meta-analysis quantifying the involvement of the dorsal medial subsystem (Yang et al., 2019).

Thank you for raising our attention to these very relevant references. We have expanded our citation related to the role of DMN in reading comprehension in this revision. Please see below for some examples.

On Page 3:

“Contemporary work in cognitive neuroscience has shown that both reading for comprehension (Andrews-Hanna, Smallwood, and Spreng, 2014; Binder, Desai, Graves, and Conant, 2009; Mar, 2011; Yang et al., 2019), and off-task states (Christoff, Gordon, Smallwood, Smith, and Schooler, 2009; Konu et al., 2020), are linked to activity within the default mode network (DMN).”

On Page 9:

“Prior studies have linked both semantic and autobiographical memory processes to the broader DMN (e.g., Andrews-Hanna, Smallwood, et al., 2014; Lambon Ralph et al., 2017; Ritchey and Cooper, 2020; Sormaz et al., 2017; Spreng, Mar, and Kim, 2009; Yang et al., 2019), and we examined how the neural patterns associated with our states reflected the activation of different subsystems of the DMN (Andrews-Hanna et al., 2010; Yeo et al., 2011).”

11. I would like to see more discussion surrounding the resting state individual difference relationships between mind-wandering during reading and connectivity with the dorsal occipital cortex. Since this region was activated in the autobiographical memory task, I suspect the authors hypothesized the opposite relationship as to that which they observed.

As we previously mentioned, we found the functional decoupling of reading-related DMN to visual cortex is associated with individual differences in mind-wandering during reading when using Reading/Autobiographical memory DMN peak seeds. This resulting pattern provides supportive evidence for our argument that mind-wandering disrupts ongoing reading comprehension since it creates a perceptually decoupled state. The results of this analysis have been updated in this revision, please see the response to point 3 for details.

12. Finally, I would like to see more detail presented on the nature of the expository text itself. Perhaps the authors could include a few example stimuli in the manuscript? Are the sentences neutral in valence? Is the text interesting to most participants, or was it meant to be boring? Also, I can't seem to find how long the duration of the memory retrieval and reading components of the trials were, on average. Could the authors add this information into the methods?

We have added a few examples of our sentence materials in the Methods section now, and yes, these sentences are neutral in valence and supposed to be boring since they described largely unfamiliar facts about each key word (e.g., “Posters are also used for reproductions of artwork, particularly famous works, and are low-cost compared to the original artwork” for the keyword POSTER). As we discussed, the type of sentence contents might also be a potential factor affecting the likelihood to which the mind-wandering occurs during reading, such as reading factual sentences vs. personally-relevant sentences, which might consequently reveal a neural pattern different from what we found using factual sentences. These open questions require future studies to address.

In addition, for each reading or memory recall trial, the duration was at a range of 26.2-29.9 s with an average of 28.0 s, and each letter string trial lasted from 13.0-15.0 s with an average of 14.0 s. We have also included the timing information of each trial in the Methods section now.

Added examples of sentences materials in the Methods section on Page 23:

“The sentences were constructed by using these key words as a search term in Wikipedia to identify text that described largely unfamiliar facts about each item; in this way, the contents of these sentences were neutral in valence (Sentence Length: Mean ± SD = 20.04 ±.93 words). For example, “Posters are also used for reproductions of artwork, particularly famous works, and are low-cost compared to the original artwork” for the keyword POSTER, and “All mammals have some hair on their skin, even marine mammals like whales and dolphins, which appear to be hairless” for the keyword MAMMAL.”

Clarification of the duration of trials in the Methods section on Page 24-25:

“Each run lasted 12.85 minutes, and each reading or memory recall trial lasted in the range of 26.2-29.9 s with an average of 28.0 s, while each letter string trial lasted from 13.0-15.0 s with an average of 14.0 s.”